# FINE-TUNING DONE *Right* IN MODEL EDITING

**Wanli Yang**[◐⚏]   **Rui Tang**[◐⚏]   **Hongyu Zang**   **Du Su**[◐]
**Qi Cao**[◐]   **Jingang Wang**   **Huawei Shen**[◐⚏]   **Xueqi Cheng**[◐⚏]   **Fei Sun**[◐✉]
[◐]State Key Laboratory of AI Safety, Institute of Computing Technology, CAS
[⚏]University of Chinese Academy of Sciences
yangwanli24z@ict.ac.cn      ✉sunfei@ict.ac.cn

## ABSTRACT

Fine-tuning, a foundational method for adapting large language models, has long been considered ineffective for model editing. Here, we challenge this belief, arguing that the reported failure arises not from the inherent limitation of fine-tuning itself, but from adapting it to the sequential nature of the editing task, a single-pass *depth-first* pipeline that optimizes each sample to convergence before moving on. While intuitive, this depth-first pipeline coupled with sample-wise updating over-optimizes each edit and induces interference across edits. Our controlled experiments reveal that simply restoring fine-tuning to the standard *breadth-first* (i.e., epoch-based) pipeline with mini-batch optimization substantially improves its effectiveness for model editing. Moreover, fine-tuning in editing also suffers from suboptimal tuning parameter locations inherited from prior methods. Through systematic analysis of tuning locations, we derive **LocFT-BF**, a simple and effective localized editing method built on the restored fine-tuning framework. Extensive experiments across diverse LLMs and datasets demonstrate that LocFT-BF outperforms state-of-the-art methods by large margins. Notably, to our knowledge, it is the first to sustain **100K** edits and **72B**-parameter models, **10 × beyond prior practice**, without sacrificing general capabilities. By clarifying a long-standing misconception and introducing a principled localized tuning strategy, we advance fine-tuning from an underestimated baseline to a leading method for model editing, establishing a solid foundation for future research.[↻]

> "*It ain't what you don't know that gets you into trouble. It's what you know for sure that just ain't so.*"
>
> — Mark Twain

## 1 INTRODUCTION

Model editing has emerged as a promising approach to efficiently update knowledge in Large Language Models (LLMs) without costly retraining (Yao et al., 2023; Wang et al., 2024d). In response, various algorithms have been explored, including parameter-extension (Hartvigsen et al., 2023; Wang et al., 2024b), meta-learning (Mitchell et al., 2022; Li et al., 2025), and locate-then-edit (Meng et al., 2022; 2023; Fang et al., 2025) methods. In contrast to these specialized approaches, direct fine-tuning, a widely recognized and effective method for adapting LLMs (Zhao et al., 2025), has nevertheless been consistently dismissed in model editing as a weak baseline, typically attributed to overfitting and catastrophic forgetting (Zhu et al., 2020). This contradiction raises a critical question: *is fine-tuning inherently unsuitable with model editing, or have we simply been using it wrong?*

In this paper, we argue that this discrepancy arises from the way it has been commonly applied in model editing studies, not the method itself. Our analysis of existing codebases reveals that fine-tuning in model editing deviates from the standard paradigm, reshaped to match the editing task where edit requests naturally arrive one by one. Concretely, rather than iterating over the entire dataset across epochs, it adopts a single-pass procedure, repeatedly optimizing each edit until fully

---

✉Corresponding author: Fei Sun (sunfei@ict.ac.cn)
[↻]Our code is available at https://github.com/ICT-STAR/LocFT

"memorized" before moving to the next. To distinguish the two, we refer to the conventional form as *fine-tuning with a **breadth-first (BF) pipeline*** and its model-editing adaptation as *fine-tuning with a **depth-first (DF) pipeline*** (typically with batch size 1). Through this lens, two inherent issues of the depth-first pipeline emerge:

**i)** its single-pass depth-first pipeline suffers catastrophic forgetting as later edits overwrite earlier ones;

**ii)** its sample-wise optimization (i.e., batch size of 1) tends to produce high-variance gradients, destabilizing the edited model's general capabilities.

To validate this hypothesis, we conduct controlled experiments on representative fine-tuning based editing methods (e.g., FT-M and AdaLoRA (Zhang et al., 2024)) via two orthogonal modifications:

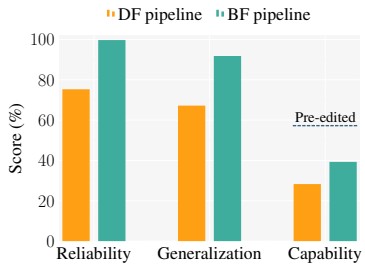

**i) Pipeline**: switching from a depth-first to a breadth-first pipeline while keeping batch size fixed at 1;

**ii) Granularity**: substituting per-sample updates (batch size = 1) with standard mini-batch optimization under the breadth-first pipeline.

As illustrated in Figure 1, our first adjustment to the optimization pipeline alone yields substantial improvements on the editing task. This suggests that the breadth-first pipeline effectively mitigates catastrophic forgetting, a long-standing weakness often criticized in fine-tuning for model editing. Building on the first step, the second controlled experiment changes only the update granularity within the breadth-first pipeline, thereby substantially reducing the degradation of general capabilities in edited models. Overall, we show that simply restoring fine-tuning based editing baselines to the standard breadth-first pipeline with mini-batch optimization yields unexpectedly strong performance on editing tasks—a result surprising at first glance, yet reasonable in hindsight.

Figure 1: Pipeline comparison of FT-M on LLaMA3-8B with 1000 ZsRE samples.

Despite these encouraging results, the fine-tuning variants we revisited still retain ad-hoc practices from prior model editing research. Specifically, they tune parameters at the locations identified by locate-then-edit methods, which are often suboptimal. This leads to a key yet underexplored question: *which layers and modules of an LLM are most effective to tune for model editing?*

To answer this question, we conduct a systematic study of tuning locations across layers and modules (e.g., attention and MLP) in diverse LLMs. Our experiments reveal that, although optimal configurations can vary across models, a general pattern emerges: tuning the down- or up-projection matrices in later layers often achieves near-perfect editing success while preserving general capabilities. Notably, this strategy remains highly effective even when not optimal.

These analyses lead to LocFT-BF (**L**ocalized **F**ine-**T**uning with **B**readth-**F**irst pipeline), a simple and effective model editing method that restores fine-tuning to its principled configuration: breadth-first pipeline, mini-batch gradient aggregation, and localized parameter updates. Unlike existing methods, LocFT-BF avoids the typical overhead of prior approaches: matrix precomputation required by locate-then-edit methods, additional labeled data required by meta-learning methods, and architectural modifications required by parameter-extension methods. This principled simplicity makes it easy to implement, efficient to run, and broadly applicable across architectures.

To evaluate the effectiveness of our method, we conduct extensive experiments on multiple representative LLMs and datasets. On the widely adopted lifelong editing task, LocFT-BF substantially outperforms state-of-the-art methods, exceeding the best baselines by an average of **33.72%** in editing success rate while consistently maintaining general capabilities of edited models. To further test its limits, we push evaluation to two extremes: **100K sequential edits** and **72B-parameter model**, both an order of magnitude beyond mainstream practice, thereby reflecting scenarios closer to real-world applications. To our knowledge, this is the first method in model editing research that can sustain 100K sequential edits while preserving general capabilities and scales efficiently with stable performance from 7B to 72B models. These results overturn the long-standing view of fine-tuning as a weak baseline and highlight its potential as a scalable solution for model editing.

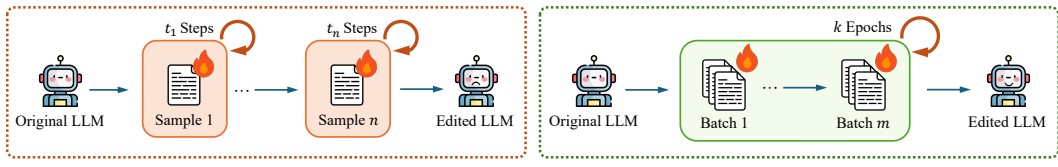

Figure 2: Illustration of edit pipeline (Depth-First) and standard pipeline (Breadth-First).

## 2 IMPLEMENTATION MATTERS IN FINE-TUNING

This section revisits the widely reported underperformance of fine-tuning in model editing (Wang et al., 2024b; Fang et al., 2025), identifying its root causes in flawed training pipeline and showing how correcting it restores fine-tuning as a competitive editing approach.

### 2.1 BACKGROUND

Model editing aims to efficiently revise specific factual knowledge in LLMs through localized parameter updates while preserving unrelated knowledge and capabilities. Formally, let $f_\theta$ denote an LLM that encodes a fact triple $t = (s, r, o)$ of subject $s$, relation $r$, and object $o$. Given a desired update $t' = (s, r, o')$ where $o' \neq o$, the editing algorithm $\mathcal{E}$ computes a parameter shift $\theta \to \theta^*$ such that the updated model $f_{\theta^*}$ predicts $o'$ when provided with the prompt $\mathrm{p}(s, r)$. For instance, to update the US presidency, $\mathcal{E}$ ensures that $f_{\theta'}$ outputs $o' = $ *Donald Trump* instead of $o = $ *Joe Biden* with prompt $\mathrm{p}(s, r) = $ *The president of the United States is*.

In real-world applications, knowledge updates emerge as a continuous stream. Consequently, the model editing task is typically formulated as a sequential process, in which the model adapts to new edits successively over time. This setting, formally known as *sequential model editing* (or *lifelong model editing*), involves a cumulative editing trajectory $f_{\theta_0} \to \cdots \to f_{\theta_k}$, where at step $k$, the edited model $f_{\theta_k}$ is required to correctly encode all $k$ target facts.

### 2.2 MIS-SPECIFIED IMPLEMENTATION

Through a detailed examination of existing fine-tuning based editing methods (Zhu et al., 2020; Wang et al., 2024a), we find that they are typically implemented following the logic of editing tasks rather than the standard fine-tuning paradigm, simulating the sequential arrival of knowledge updates. Specifically, they adopt a sample-by-sample training procedure: repeatedly optimizing each sample to convergence before advancing to the next. We refer to this approach as a *Depth-First (DF) pipeline*. In contrast, as illustrated in Figure 2, the standard *Breadth-First (BF) pipeline* differs in two key aspects: ❶ it iterates over the entire dataset across epochs and ❷ employs mini-batch updates rather than sample-wise optimization. While the DF design appears intuitive in the context of editing tasks, it is not a necessary choice in practice: real-world edits rarely arrive strictly sample by sample, and even when updates are sequential, the BF pipeline can incorporate them incrementally. More importantly, the DF pipeline inherently introduces two risks: ❶ the sequential and single-pass nature makes it prone to later edits overwriting earlier ones; ❷ the sample-wise optimization tends to produce unstable gradients, which can destabilize the general capabilities of edited models.

### 2.3 IMPACT OF TRAINING PIPELINE

To validate our hypothesis regarding pipelines, we compare DF and BF in controlled experiments, fixing batch size at 1 and keeping other settings constant. We evaluate four representative fine-tuning based methods: FT-L (Zhu et al., 2020), FT-M (Zhang et al., 2024), AdaLoRA (Zhang et al., 2023), and RoseLoRA (Wang et al., 2024a), on two mainstream datasets: ZsRE (Levy et al., 2017) and COUNTERFACT (Meng et al., 2022), using three popular LLMs: LLaMA-3-8B-Instruct (Grattafiori et al., 2024), Mistral-7B-v0.1 (Jiang et al., 2023), and Qwen2.5-7B-Instruct (Qwen et al., 2025). Performance is measured with three metrics: *Reliability*, *Generalization*, and *Capability*. Further details of the methods, datasets, LLMs, and metrics are provided in § 4.1 and Appendix A.1.

Table 1: Performance comparison of DF and BF pipelines on mainstream fine-tuning based editors.

| | Method | ZsRE | | | | | | COUNTERFACT | | | | | |
|---|---|---|---|---|---|---|---|---|---|---|---|---|---|
| | | Reliability | | Generalization | | Capability | | Reliability | | Generalization | | Capability | |
| | | DF | BF | DF | BF | DF | BF | DF | BF | DF | BF | DF | BF |
| **LLaMA** | FT-L | 0.00 | 0.00 | 0.00 | 0.10 | 14.96 | 23.47 | 0.00 | 0.00 | 0.00 | 0.00 | 15.14 | 18.17 |
| | FT-M | 75.30 | 99.70 | 67.20 | 91.80 | 28.30 | 39.30 | 80.00 | 99.90 | 52.60 | 75.10 | 29.89 | 30.87 |
| | AdaLoRA | 3.80 | 90.50 | 3.30 | 69.70 | 44.81 | 49.89 | 8.40 | 96.30 | 6.10 | 44.10 | 49.88 | 39.27 |
| | RoseLoRA | 0.30 | 1.00 | 0.00 | 0.70 | 57.13 | 57.38 | 0.30 | 0.20 | 0.00 | 0.00 | 57.17 | 56.66 |
| **Mistral** | FT-L | 0.00 | 0.00 | 0.00 | 0.00 | 15.73 | 15.38 | 0.00 | 0.00 | 0.00 | 0.00 | 14.93 | 15.34 |
| | FT-M | 41.10 | 100.00 | 24.60 | 83.50 | 18.14 | 15.64 | 59.70 | 99.90 | 25.60 | 59.10 | 16.94 | 20.50 |
| | AdaLoRA | 2.50 | 88.50 | 2.40 | 74.70 | 25.36 | 41.27 | 5.10 | 95.50 | 3.90 | 44.70 | 19.14 | 39.63 |
| | RoseLoRA | 0.20 | 4.10 | 0.00 | 3.60 | 45.24 | 43.80 | 0.60 | 0.80 | 0.20 | 0.20 | 45.57 | 43.97 |
| **Qwen** | FT-L | 0.10 | 0.10 | 0.00 | 0.10 | 23.65 | 29.66 | 0.10 | 0.60 | 0.10 | 0.50 | 31.21 | 33.07 |
| | FT-M | 58.70 | 99.80 | 34.60 | 77.60 | 25.41 | 34.28 | 67.70 | 99.90 | 25.60 | 35.40 | 32.57 | 33.17 |
| | AdaLoRA | 3.40 | 90.60 | 2.50 | 54.00 | 53.47 | 51.21 | 8.90 | 97.00 | 4.00 | 19.10 | 41.74 | 53.23 |
| | RoseLoRA | 0.00 | 0.40 | 0.00 | 0.10 | 58.73 | 58.58 | 0.20 | 0.40 | 0.10 | 0.10 | 58.48 | 58.37 |

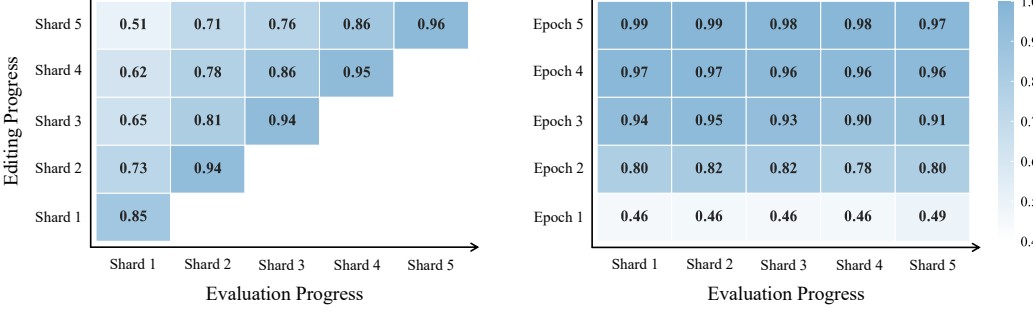

Figure 3: Visualization of the learning dynamics for depth-first and breadth-first pipelines.

Table 1 shows that replacing the DF pipeline with BF improves performance across all methods, with especially large gains for FT-M and AdaLoRA. FT-L and RoseLoRA show only minor improvements due to their design constraints: FT-L optimizes only on the last token of the target answer, while RoseLoRA excessively limits the scope of trainable parameters. Overall, these results highlight the critical role of a proper training pipeline in effective knowledge editing.

Although transitioning from DF to BF pipeline yields substantial gains, especially in editing success, the mechanism behind this gap remains unclear. To investigate this, we compare the learning dynamics of the two pipelines using FT-M on LLaMA-3-8B-Instruct with 1000 ZsRE samples. Specifically, we randomly partition the 1000 samples into five equal shards and track the average editing success of each shard during learning. For DF pipeline, the shards are edited sequentially; and after completing each shard we evaluate on all previously edited ones to detect potential overwriting. While for BF pipeline, all samples are shuffled each epoch, and after every epoch we evaluate each shard separately.

The visualization results are depicted in Figure 3. Under DF pipeline, earlier shards start with high success rates but decline as later shards are edited. In contrast, under BF pipeline, all shards improve jointly across epochs and converge to near-perfect success rates. These results confirm our hypothesis that the mis-specified DF pipeline induces catastrophic overwriting of earlier edits.

## 2.4 IMPACT OF GRADIENT AGGREGATION

While fixing the training pipeline markedly improves editing success and generalization, the edited models still show clear degradation in general capabilities, especially for FT-M. We next validate the second hypothesized drawback: per-sample updates (batch size = 1) destabilize the edited model. To test this, we replace per-sample updates with mini-batch training within each epoch for both FT-M and AdaLoRA, further aligning fine-tuning–based editing with standard practice.

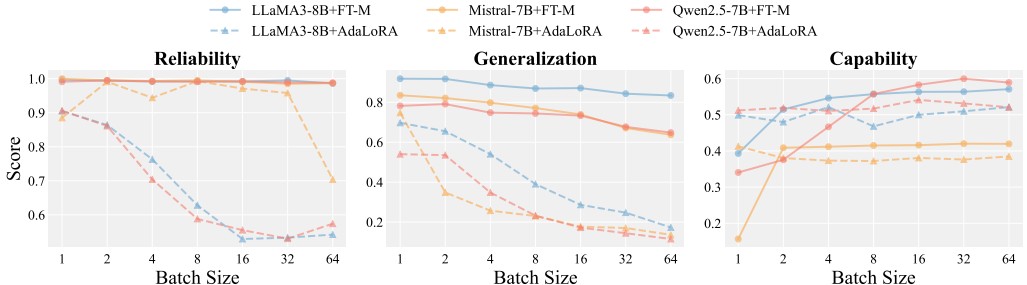

Figure 4: Impact of batch size on editing performance under the BF pipeline on ZsRE.

As shown in Figure 4, increasing the batch size substantially improves the capability performance of FT-M to even surpass AdaLoRA, indicating that mini-batch training stabilizes model states during editing. Conversely, AdaLoRA attains modest gains due to its low-rank design, which already regularizes updates and preserves downstream performance. While larger batches reduce generalization to some extent, more noticeably for AdaLoRA and slightly for FT-M, this effect can be readily mitigated through data augmentation (Gangadhar & Stratos, 2024), given that each edit currently relies on only a single QA pair. Similar observations regarding the negative impact of large batch sizes on generalization have also been reported by Keskar et al. (2017).

## 2.5 Breadth-first Pipeline for Sequential Editing

To address sequential editing in real-world scenarios, the BF pipeline offers flexible adaptation strategies. In strictly real-time settings, each incoming edit is immediately integrated into the cumulative dataset to trigger a new fine-tuning process. In practical deployments where latency constraints are relaxed, efficiency can be further optimized by buffering new requests into small batches before updating. Both strategies are highly feasible, given that the fine-tuning process is lightweight (as shown in Section 4.2) and the editing workload is negligible compared to pre-training corpora.

In summary, aligning fine-tuning based editing with the standard breadth-first, mini-batch paradigm repositions it from a perceived weak baseline to a competitive approach. This finding rectifies a long-standing misconception: the widely reported failure arose not from the inherent limitations of fine-tuning, but from the ill-suited implementations in prior work.

## 3 Tailoring Fine-tuning for Model Editing

Our studies demonstrate that transitioning to a standard fine-tuning pipeline substantially improves editing performance. Nevertheless, existing fine-tuning variants remain coarse and insufficiently designed, typically tuning the location identified by locate-then-edit methods without theoretical or empirical support, resulting in suboptimal generalization and capability. This raises a critical question: *which locations of an LLM are most effective to tune for model editing?*

To address it, we conduct a comprehensive study of tuning locations to establish a rigorous basis for parameter selection in fine-tuning based editing. Specifically, we examine the same three LLMs as in earlier experiments, along two dimensions: *layer* and *module*. For each layer, we test five candidate modules: entire layer, full attention, full MLP, $MLP_{up}$, and $MLP_{down}$. This design is motivated by different emphases in the literature: parameter-efficient fine-tuning (Hu et al., 2022) typically targets the attention modules for downstream adaptation, whereas editing (Meng et al., 2022) and mechanistic studies (Geva et al., 2021) highlight the MLP, especially $MLP_{down}$, as the locus of factual knowledge. For example, this yields 160 tuning locations in LLaMA3-8B (32 layers $\times$ 5 modules), each fine-tuned independently. We also evaluated full-parameter and multi-layer fine-tuning; however, their poor performance and disruption of general capabilities led us to exclude them.

For the editing setup, we perform localized fine-tuning on 1000 ZsRE samples and evaluate three metrics: *Reliability*, *Generalization*, and *Capability* (assessed on three representative tasks, GSM8K, MMLU, and WMT16, to reduce resource cost). Corresponding results are presented in Figure 5.

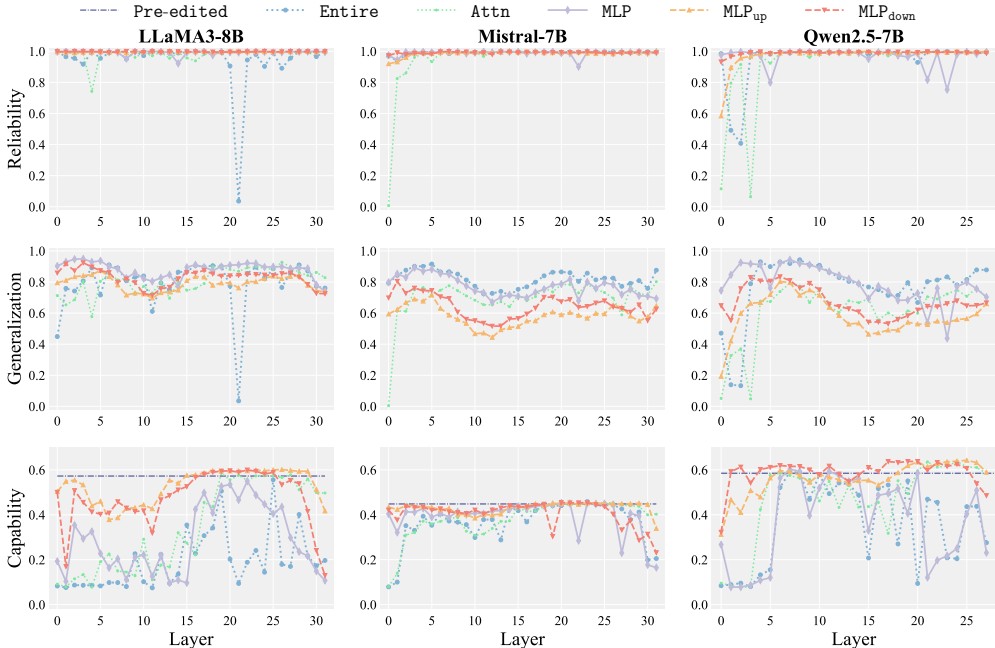

Figure 5: Fine-tuning performance for different locations across three LLMs.

**Reliability Perspective**. Across all models, fine-tuning nearly any module in any layer achieves near-perfect editing success, suggesting that knowledge acquisition capacity is broadly distributed rather than confined to specific parameters. This is consistent with recent studies challenging knowledge localization (Chen et al., 2025b) and showing that both attention and MLP modules can encode knowledge (Wei et al., 2024a; Li et al., 2024). However, tuning the entire layer, full attention, or full MLP shows slight instability, with occasional drops at certain layers. In contrast, $\text{MLP}_{\text{down}}$ is the most stable, consistently maintaining high reliability across layers, consistent with prior findings that it is particularly specialized for knowledge update (Geva et al., 2021; Meng et al., 2022).

**Generalization Perspective**. Unlike the uniformly high reliability, generalization varies notably across layers and modules, though the overall pattern is consistent across all three LLMs. *From the module perspective*, coarser-grained components such as entire layer and full MLP generalize better than finer-grained counterparts like $\text{MLP}_{\text{up}}$ and $\text{MLP}_{\text{down}}$, suggesting broader parameter updates may facilitate more robust memorization and stronger generalization to paraphrased prompts. *From the layer perspective*, generalization follows a consistent trajectory: it peaks in early layers (e.g., layers 3–8), declines in middle layers, shows a smaller secondary peak in later layers (e.g., layers 20–25), and then declines again. This highlights the non-trivial impact of layer choice and the need for systematic empirical assessment in guiding effective selection.

**Capability Perspective.** Unlike generalization, which favors coarser components, capability is stronger and more stable with fine-grained modules, i.e., $\text{MLP}_{\text{up}}$ and $\text{MLP}_{\text{down}}$. This shows that more localized updates cause less disruption, helping edited models preserve general capability. *From the layer perspective*, capability follows a trajectory similar to generalization, with peaks in early and later layers. However, unlike generalization's preference for the early layers, capability is more pronounced in the later layers, further underscoring their trade-off.

Overall, since most locations yield high reliability, tuning location selection mainly depends on the trade-off between generalization and capability. We prioritize capability for two reasons: ❶ preserving LLMs' general abilities is essential for editing; ❷ generalization, even if suboptimal, can be improved with data augmentation (Gangadhar & Stratos, 2024), as current editing uses only one QA pair per fact. Based on this and

Table 2: Selected tuning locations.

| Model | Tuning Location |
|---|---|
| LLaMA3-8B | $\text{MLP}_{\text{down}}$ in layer 22 |
| Mistral-7B | entire layer 23 |
| Qwen2.5-7B | $\text{MLP}_{\text{down}}$ in layer 6 |

our experiments, we identify the relatively optimal tuning locations for each LLM, as in Table 2. Besides, the results reveal a consistent pattern: editing $\text{MLP}_{\text{down}}$ in the later layers minimizes capability degradation while retaining near-perfect reliability and acceptable generalization. Notably,

this strategy remains highly effective even when not optimal. For instance, although Qwen performs best in earlier layers, later layers still achieve strong results. We further validate the universality of this strategy in Appendix A.2, demonstrating that it generalizes seamlessly and achieves robust performance on Mixture-of-Experts (MoE) models and unseen QAEdit dataset (Yang et al., 2025).

Integrating three key factors, namely breadth-first pipeline, mini-batch gradient aggregation, and tuning-location selection, yields **LocFT-BF**, a simple yet powerful fine-tuning method for model editing. Owing to the inherent simplicity and extensibility of fine-tuning, LocFT-BF is broadly applicable, plug-and-play, and avoids the overhead of locate-then-edit (matrix precomputation) and meta-learning (extra training). Although LocFT-BF requires specifying tuning locations, our study offers an initial, relatively stable selection strategy to address this.

## 4 BENCHMARKING LOCFT-BF IN LIFELONG EDITING

To position LocFT-BF against state-of-the-art editing methods, we conduct a comprehensive evaluation across diverse LLMs and datasets under a lifelong editing setup.

### 4.1 EXPERIMENTAL SETUP

**Editing Methods.** To ensure comprehensive coverage, we compare LocFT-BF against six representative editing methods across three categories: parameter-extension (**WISE**, Wang et al., 2024b), meta-learning (**RLEdit**, Li et al., 2025 and **UltraEdit**, Gu et al., 2025), and locate-then-edit (**MEMIT**, Meng et al., 2023, **RECT**, Gu et al., 2024, and **AlphaEdit**, Fang et al., 2025).

**Edited LLMs.** Following prior work (Fang et al., 2025; Gu et al., 2025), we evaluate three leading open-source LLMs: LLaMA-3-8B-Instruct (Grattafiori et al., 2024), Mistral-7B-v0.1 (Jiang et al., 2023), and Qwen2.5-7B-Instruct (Qwen et al., 2025).

**Editing Datasets.** In line with prior studies (Wang et al., 2024b; Fang et al., 2025), we randomly sample 3000 instances from ZsRE (Levy et al., 2017), COUNTERFACT (Meng et al., 2022), and WikiBigEdit (Thede et al., 2025) for large-scale editing evaluation. Beyond these mainstream factual editing benchmarks, we further evaluate the applicability of editing methods on a medical editing benchmark, MedEditBench (Chen et al., 2025a), with details provided in Appendix A.3.

**Evaluation Metrics.** We evaluate editing techniques from four key properties: ❶ *Reliability*: success rate of editing; ❷ *Generalization*: adaptability of edited knowledge to rephrased prompts; ❸ *Capability*: preservation of general capabilities, measured as the average accuracy of edited models on MMLU (Hendrycks et al., 2021), Natural Questions (Kwiatkowski et al., 2019), SST2 (Socher et al., 2013), WMT16 (Bojar et al., 2016), and GSM8K (Cobbe et al., 2021); ❹ *Efficiency*: average time to perform each edit. Notably, we evaluate *Reliability* and *Generalization* using the WILD framework (Yang et al., 2025), which employs autoregressive decoding instead of conventional teacher forcing generation to align with practical deployment scenario and avoid overestimation.

Detailed information regarding method implementation, LLM configuration, dataset preparation, and evaluation procedures is provided in Appendix A.1.

### 4.2 RESULTS & ANALYSIS

The results are reported in Table 3. To structure the analysis, we examine them from four evaluation perspectives: reliability, generalization, capability, and efficiency.

**Reliability.** Remarkably, LocFT-BF consistently attains the highest reliability across all nine dataset–LLM combinations, with absolute gains of **33.72%** on average and up to **58.50%** over the second-best. This demonstrates that fine-tuning can achieve effective knowledge updates. Although recent methods such as AlphaEdit, RLEdit, and UltraEdit achieve relatively strong reliability compared to traditional approaches, they exhibit notable instability across LLMs and datasets.

**Generalization.** LocFT-BF exhibits superior generalization, achieving optimal performance in six out of nine dataset–LLM combinations. The two least effective cases are from the COUNTERFACT dataset, whose generalization prompts are constructed by prepending irrelevant text rather than direct paraphrasing, which makes generalization particularly challenging. Methods like MEMIT and

Table 3: Comparison of LocFT-BF with existing methods on lifelong editing task. The best results are denoted in **bold** and the second-best results are underlined.

| Data | Method | LLaMA3-8B | | | | Mistral-7B | | | | Qwen2.5-7B | | | |
|---|---|---|---|---|---|---|---|---|---|---|---|---|---|
| | | Rel. | Gen. | Cap. | Time | Rel. | Gen. | Cap. | Time | Rel. | Gen. | Cap. | Time |
| | Pre-edited | – | – | 57.26 | – | – | – | 44.82 | – | – | – | 58.52 | – |
| ZsRE | MEMIT | 26.23 | 23.30 | 25.67 | 9.90 | 24.30 | 19.60 | 18.11 | 10.28 | 39.57 | 32.10 | 47.87 | 9.93 |
| | RECT | 0.03 | 0.03 | 14.98 | 25.38 | 0.17 | 0.27 | 14.88 | 25.79 | 9.70 | 8.20 | 16.06 | 24.31 |
| | WISE | 4.17 | 3.50 | – | 14.42 | 15.87 | 11.33 | – | 13.02 | 7.67 | 5.23 | – | 30.96 |
| | AlphaEdit | 64.50 | 40.57 | 54.71 | 12.31 | 3.30 | 3.00 | 15.13 | 10.95 | 2.70 | 2.33 | 16.31 | 10.55 |
| | RLEdit | 66.67 | 59.40 | **57.41** | 0.58 | 26.10 | 19.53 | 21.30 | 0.47 | 54.57 | 47.60 | **60.74** | 0.65 |
| | UltraEdit | 34.93 | 22.67 | 55.93 | **0.22** | 40.43 | 23.40 | 44.22 | **0.04** | 40.10 | 19.77 | 59.25 | **0.27** |
| | **LocFT-BF** | **98.97** | **75.83** | 57.04 | 0.27 | **98.93** | 49.57 | 42.73 | 0.31 | **98.87** | 74.37 | 59.07 | 0.49 |
| COUNTERFACT | MEMIT | 71.90 | **48.47** | 19.30 | 9.34 | 37.83 | **28.17** | 15.59 | 9.06 | 68.37 | **40.90** | 44.00 | 8.70 |
| | RECT | 0.53 | 0.13 | 14.85 | 21.68 | 0.77 | 0.37 | 15.67 | 22.21 | 0.00 | 0.00 | 14.81 | 21.69 |
| | WISE | 19.80 | 13.13 | – | 12.18 | 25.13 | 4.60 | – | 10.28 | 20.17 | 10.20 | – | 27.18 |
| | AlphaEdit | 94.27 | 39.90 | 54.09 | 10.60 | 6.67 | 6.70 | 15.47 | 9.75 | 32.17 | 18.10 | 17.46 | 9.46 |
| | RLEdit | 65.33 | 33.23 | 55.45 | 0.46 | 36.30 | 17.07 | 23.53 | 0.40 | 44.33 | 18.90 | 58.48 | 0.45 |
| | UltraEdit | 68.33 | 31.20 | 56.85 | **0.18** | 57.60 | 22.60 | **44.91** | **0.03** | 41.33 | 15.33 | 59.01 | **0.18** |
| | **LocFT-BF** | **99.73** | 33.23 | **57.13** | 0.38 | **99.67** | 39.53 | 41.46 | 0.24 | **99.73** | 11.77 | 58.53 | 0.48 |
| WikiBigEdit | MEMIT | 10.77 | 10.80 | 23.57 | 10.39 | 12.47 | 10.07 | 18.26 | 11.02 | 31.03 | 25.27 | 47.92 | 10.69 |
| | RECT | 0.00 | 0.00 | 14.90 | 30.43 | 0.00 | 0.00 | 14.81 | 30.41 | 1.87 | 0.97 | 14.78 | 27.96 |
| | WISE | 30.33 | 27.03 | – | 15.14 | 38.03 | 32.53 | – | 11.50 | 34.40 | 30.63 | – | 33.80 |
| | AlphaEdit | 64.73 | 51.17 | 54.14 | 14.18 | 3.37 | 3.57 | 14.69 | 12.05 | 4.83 | 3.67 | 15.13 | 11.68 |
| | RLEdit | 71.10 | 63.27 | 57.28 | 0.60 | 60.83 | 46.47 | 36.20 | 0.38 | 66.40 | 58.50 | 59.36 | 0.47 |
| | UltraEdit | 72.67 | 65.37 | **57.98** | 0.20 | 33.87 | 27.00 | 25.43 | **0.05** | 74.20 | **59.17** | 58.82 | **0.31** |
| | **LocFT-BF** | **98.87** | **73.77** | 56.93 | 0.54 | **98.90** | **74.97** | **42.39** | 0.39 | **99.43** | 53.30 | **59.83** | 0.98 |

AlphaEdit explicitly enhance robustness to such noise through data augmentation, accounting for their relative advantage over LocFT-BF on this dataset. In Appendix A.4, we demonstrate that by incorporating the same data augmentation strategy, LocFT-BF achieves substantial gains in generalization, effectively surpassing both MEMIT and AlphaEdit.

**Capability.** LocFT-BF and recently proposed methods, including AlphaEdit, RLEdit, and UltraEdit, exhibit advantages in preserving the general capabilities of edited models under large-scale editing. However, these baseline methods show noticeable instability across LLMs: RLEdit and UltraEdit struggle on Mistral-7B, while AlphaEdit, despite performing well on LLaMA3-8B, fails to generalize to the other two LLMs. In contrast, LocFT-BF demonstrates the strongest stability, consistently maintaining high capability across all evaluated LLMs and datasets. This highlights the key strength of fine-tuning: its simple and general design makes it broadly adaptable across architectures.

**Efficiency.** LocFT-BF, RLEdit, and UltraEdit achieve top-tier efficiency, completing each edit within one second by directly updating model parameters. Conversely, other baselines are nearly 50 times slower, primarily due to their more complex procedures, such as auxiliary module optimization (WISE) and additional matrix calculation (MEMIT, RECT, and AlphaEdit), which introduce substantial computational overhead.

In summary, LocFT-BF delivers superior performance across all key evaluation dimensions while maintaining strong stability, establishing it as an effective and robust technique for model editing. Notably, we also implement our method within **LlamaFactory** (Zheng et al., 2024), observing comparable performance with significantly enhanced efficiency, as detailed in Appendix A.5.

## 5 SCALING TOWARDS REAL-WORLD EDITING

We further extend our evaluation to more realistic scenarios by scaling data volume and model size to 10× the scale of mainstream practice, thereby probing the limits of our approach.

### 5.1 SCALING TO LARGER DATA VOLUMES

Owing to the limited capacity of existing editing techniques, prior evaluations have typically been restricted to approximately 3000 edits. However, such a scale is insufficient to reflect real-world requirements for lifelong editing. To address the limitation, we scale the number of edits to better reflect practical scenarios.

Table 4: Editing performance across Qwen2.5 models as scale increases from 7B to 72B parameters.

| Strategy | Qwen2.5-7B | | | Qwen2.5-14B | | | Qwen2.5-32B | | | Qwen2.5-72B | | |
|---|---|---|---|---|---|---|---|---|---|---|---|---|
| | Rel. | Gen. | Cap. | Rel. | Gen. | Cap. | Rel. | Gen. | Cap. | Rel. | Gen. | Cap. |
| Pre-edited | - | - | 58.52 | - | - | 62.56 | - | - | 63.72 | - | - | 64.84 |
| Default Pos. | 99.20 | 83.40 | 59.33 | 99.00 | 78.80 | 61.62 | 99.30 | 70.00 | 60.27 | 98.60 | 65.30 | 63.64 |
| Proportional Pos. | 99.20 | 83.40 | 59.33 | 99.50 | 77.90 | 61.12 | 99.40 | 68.20 | 64.72 | 99.30 | 68.30 | 63.87 |

Figure 6: Evolution of editing performance during the scaling process to 100K edits.

**Experimental Setup.** We increase the number of edits from 3K to 100K using the ZsRE dataset. For this experiment, we evaluate LocFT-BF, RLEdit, and UltraEdit, the only three methods that maintain model capabilities under 3K sequential edits, on LLaMA3-8B and Qwen2.5-7B.

**Experimental Results.** As shown in Figure 6, LocFT-BF exhibits significant superiority over the leading baselines across all metrics when scaled to large edit volumes. Even at 100K edits, it maintains near-perfect success rates while preserving the general capabilities of edited models. To our knowledge, it is the only method to achieve both effective knowledge updates and capability retention at this scale. Although its generalization declines, LocFT-BF still substantially surpasses the baselines, and this aspect can be further improved with data augmentation. In contrast, RLEdit and UltraEdit struggle to scale, with model capabilities decline sharply when edits exceed 20K. More critically, their consistently low editing success rates reveal a fundamental limitation in achieving reliable knowledge updates.

## 5.2 SCALING TO LARGER MODELS

The complex designs of most existing model editing techniques lead to heavy computational overhead, which has confined mainstream evaluations to 7B-level LLMs. For example, locate-then-edit approaches such as ROME and AlphaEdit require computing large covariance and projection matrices, incurring substantial memory and time costs that make scaling to larger models practically infeasible. In contrast, fine-tuning is inherently lightweight and easily extensible, allowing us to push beyond this long-standing 7B barrier and evaluate LocFT-BF on much larger models, reflecting practical deployment scenarios.

**Experimental Setup.** We apply LocFT-BF to edit a series of Qwen2.5 models ranging from 7B to 72B. We evaluate with 1,000 ZsRE samples to balance evaluation coverage and computational feasibility for large models. Considering an exhaustive search for the optimal tuning position in larger models, such as Qwen2.5-72B, is computationally prohibitive, we design two heuristic strategies to determine the target layer, based on the optimal position identified in Qwen2.5-7B. ❶ Default Position: we directly apply the same absolute position (i.e., $\text{Layer6.MLP}_{down}$) to all larger models, serving as a straightforward baseline. ❷ Proportional Position: preserve the relative depth of the target layer to account for increasing model depth. For instance, since Layer 6 is the 7th layer in the 28-layer Qwen2.5-7B, we select the corresponding 20th layer (i.e., $\text{Layer19.MLP}_{down}$) for the 80-layer Qwen2.5-72B, corresponding to $80 \times (7/28) = 20$.

**Experimental Results.** The results in Table 4 demonstrate that LocFT-BF readily scales to larger models, achieving reliable knowledge updates while maintaining general capabilities. Notably, both heuristic strategies for tuning position selection prove effective, eliminating the need for an exhaustive and computationally expensive position search. For mid-sized models such as Qwen2.5-14B, the Default Position strategy is sufficient for strong performance, whereas for larger models like Qwen2.5-32B, the Proportional Position strategy yields better results. These results highlight that LocFT-BF can be seamlessly extended from 7B to 72B models without redesign, underscoring its simplicity and plug-and-play scalability in practical deployment. In contrast, scaling baseline methods to larger models remains challenging. Even the most competitive methods, RLEdit and UltraEdit, exhibit further performance degradation on 14B models, as detailed in Appendix A.6.

Finally, to bridge the dimensions of data and model scaling, we evaluate LocFT-BF in a industrial-scale scenario: **editing Qwen2.5-72B with 100K samples from ZsRE**. In this challenging setting, our method achieves near-perfect reliability (**99.80%**) and robust generalization (**57.60%**), while effectively retaining the model's general capabilities (**62.54%** compared to the original 64.84%), confirming its practicality for real-world deployment.

## 6 RELATED WORKS

**Model Editing Methodologies.** Existing approaches to model editing can be broadly categorized into three groups. ❶ *Parameter-extension.* These methods introduce additional trainable components decoupled from pretrained parameters to encode new knowledge. Representative designs include extra neurons (T-Patcher, Huang et al., 2023), codebooks (GRACE, Hartvigsen et al., 2023), and auxiliary memory modules (WISE, Wang et al., 2024b). ❷ *Meta-learning.* KE (De Cao et al., 2021) and MEND (Mitchell et al., 2022) train a hypernetwork to predict parameter updates for knowledge editing. RLEdit (Li et al., 2025) further enhances this approach with reinforcement learning, enabling the hypernetwork to adaptively produce parameter updates conditioned on the edited model's evolving state. ❸ *Locate-then-edit.* Building upon research into knowledge mechanisms of LLMs (Geva et al., 2021; 2022), ROME (Meng et al., 2022) and MEMIT (Meng et al., 2023) apply causal tracing to identify knowledge-critical parameters and apply localized updates for precise editing. AlphaEdit (Fang et al., 2025) augments this line by projecting parameter updates onto the null space of preserved knowledge to mitigate disruption in lifelong editing.

**Fine-tuning Based Model Editing.** FT-L (Zhu et al., 2020; Meng et al., 2022) and FT-M (Zhang et al., 2024) directly apply fine-tuning to parameters identified by the locate-then-edit paradigm but suffer from catastrophic forgetting. To mitigate this limitation, recent studies have focused on two primary strategies. ❶ *Parameter-efficient fine-tuning.* MELO (Yu et al., 2024) and RoseLoRA (Wang et al., 2024a) incorporate low-rank adaptation into model editing to constrain the interference induced by fine-tuning. ❷ *Data augmentation.* Since prior methods rely on a single QA pair per fact, recent work (Gangadhar & Stratos, 2024; Wei et al., 2024b) proposes to enrich the editing data, thus improving the acquisition of new knowledge during fine-tuning.

In contrast to previous research that focused on architectural refinements or data augmentation, our study is, to the best of our knowledge, the first to revisit the failure of fine-tuning-based model editing, We attribute this failure to a flawed implementation. By correcting this and customizing tuning locations, we have shown that fine-tuning can be an effective paradigm, disproving the misconception that it is unsuitable for model editing.

## 7 CONCLUSION

In this paper, we present the first re-examination of fine-tuning based model editing, a technique long regarded as a weak baseline. By identifying and correcting critical flaws in existing implementations, we unlock the true potential of fine-tuning and overturn the prevailing misconception of its limitations. Building on this, we propose LocFT-BF, a pure localized fine-tuning approach grounded in comprehensive empirical analysis of tuning positions. Extensive experiments demonstrate that LocFT-BF not only significantly surpasses state-of-the-art editing methods but also scales effectively to massive editing workloads (100K samples) and large models (up to 72B parameters). This work redefines the role of fine-tuning in model editing, establishing it as a powerful and practical technique, while pushing the field closer to real-world deployment.

## ACKNOWLEDGMENTS

This work was supported by the Beijing Natural Science Foundation (4252023), the Strategic Priority Research Program of the Chinese Academy of Sciences (XDB0680201).

## ETHICS STATEMENT

This paper adheres to the ICLR Code of Ethics. All data used in our research are publicly available and contain no personally identifiable or sensitive information, thereby posing no privacy concerns. While the proposed editing method LocFT-BF has the ability to modify the knowledge of LLMs and thus carries potential risks of introducing false or harmful information, its intended purpose is positive, aiming to correct misinformation and bias in pretrained LLMs. We therefore urge researchers to apply this technology responsibly and with due caution.

## REPRODUCIBILITY STATEMENT

To ensure the reproducibility of our findings, we have made the source code open-source and documented detailed experimental settings and configurations in Appendix A.1.

## LLM USAGE

We employ LLMs to polish the manuscript, focusing on correcting grammatical errors and enhancing clarity, rather than generating new content or ideas. The authors take full responsibility for all the content of the paper, including any text refined by the LLMs, and we ensure that the usage of LLMs adheres to the ethical guidelines.

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

# A  APPENDIX

## A.1  DETAILED EXPERIMENTAL SETUP

### A.1.1  EDITING METHODS

**FT-L** (Zhu et al., 2020; Meng et al., 2022) fine-tunes the MLP of a specific layer identified through causal tracing in ROME (Meng et al., 2022), while augmenting the objective with an $l_\infty$-norm regularization that explicitly limits parameter deviations between the original and edited models to mitigate side effects on unrelated knowledge and capabilities. However, FT-L deviates from the standard fine-tuning paradigm by leveraging only the last token of the target answer as supervision, which severely undermines its editing success.

**FT-M** (Zhang et al., 2024) addresses the supervision limitation of FT-L by applying a cross-entropy loss over the entire target answer while masking the prompt, thereby aligning more closely with the standard fine-tuning practices and yielding substantial gains in editing success.

**AdaLoRA** (Zhang et al., 2023) enhances vanilla Low-Rank Adaptation (LoRA) by adaptively allocating the parameter budget among weight matrices according to their importance score. Zhang et al. (2024) directly adapt this technique to model editing and report competitive results.

**RoseLoRA** (Wang et al., 2024a) is a novel parameter-efficient fine-tuning (PEFT) method that introduces row and column-wise sparsity on the product of low-rank matrices. This approach allows it to selectively update only the most critical parameters of a pretrained language model, ensuring efficient and precise updates while preserving the irrelevant knowledge.

**WISE** (Wang et al., 2024b) targets the lifelong editing task with a dual-memory architecture composed of a main memory for pretrained knowledge, a side memory for edited knowledge, and a router that directs queries between them. By explicitly decoupling edited knowledge from pretrained knowledge, WISE effectively mitigates interference with the original model.

**RLEdit** (Li et al., 2025) reformulates hypernetwork-based lifelong editing as a reinforcement learning problem, where target edits and the state of the edited model constitute the environment, editing losses serve as rewards, and the hypernetwork is optimized at the sequence level as the policy. This design enables the hypernetwork to precisely track parameter changes in LLMs during editing and generate more accurate updates for lifelong knowledge editing.

**UltraEdit** (Gu et al., 2025) proposes a training-, subject-, and memory-free editing framework that computes parameter shifts through lightweight linear algebra operations, enabling fast and consistent updates with minimal overhead. To support lifelong adaptation, it further employs a lifelong normalization strategy that continually updates feature statistics, allowing the model to adapt to distributional shifts while preserving consistency over time.

**MEMIT** (Meng et al., 2023) extends ROME by scaling its mechanism from single-layer to multi-layer editing. It first identifies knowledge-relevant layers and modules through causal tracing analysis and then applies rank-one matrix update at the identified locations to perform target edits. By propagating modifications across multiple successive layers, MEMIT enables efficient batch editing of large-scale knowledge.

**RECT** (Gu et al., 2024) builds on ROME by introducing a regularization strategy to mitigate overfitting and reduce noise induced by editing. Specifically, it quantifies the importance of each weight element by the absolute value of its relative change and updates only the top-$k\%$ elements. This selective update effectively suppresses side effects while preserving editing performance.

**AlphaEdit** (Fang et al., 2025) augments locate-then-edit methods by projecting parameter changes onto the null space of preserved knowledge. This projection is theoretically proven to keep the outputs of post-edited LLMs unchanged when queried about preserved knowledge, thereby mitigating the issue of disruption.

To ensure a fair and rigorous comparison, we standardize the method implementations in two specific aspects. First, to maximize the editing efficacy of each method within available computational resources, we adopt an empirically tuned batch size while preserving all other algorithmic configurations from their official implementations. Second, we standardize the target answer formatting. While previous model editing implementations omit the **end-of-sequence (EOS) token**, typically

causing irrelevant or incorrect generation after the correct answer (Yang et al., 2025), standard fine-tuning practices strictly require it (Zheng et al., 2024; Sheng et al., 2025). Consequently, we append an EOS token to the target answers across all methods to ensure proper generation termination.

### A.1.2   EDITED LLMS

**LLaMA-3-8B-Instruct** (Grattafiori et al., 2024) is a leading 8-billion-parameter instruction-tuned model from the LLaMA family of Meta AI. It is designed primarily for dialogue applications and outperform many existing open-source chat models on common industry benchmarks. In addition, it has been further optimized to enhance both helpfulness and safety.

**Mistral-7B-v0.1** (Jiang et al., 2023) is a resource-efficient yet highly capable foundation model with 7 billion parameters. It consistently outperforms LLaMA-2-13B across all evaluated benchmarks and even surpasses LLaMA-1-34B on tasks involving reasoning, mathematics, and code generation.

**Qwen2.5-7B-Instruct** (Qwen et al., 2025) is a superior instruction-tuned model with 7 billion parameters, trained on 18 trillion tokens with over 1M supervised finetuning samples and multistage reinforcement learning. It demonstrates strong performance across language understanding, reasoning, mathematics, and coding, offering competitive capability while being resource-efficient.

All models adopt greedy decoding for generation, consistent with mainstream practice in model editing (Yao et al., 2023; Wang et al., 2024c).

### A.1.3   EDITING DATASETS

**ZsRE** (Levy et al., 2017) is a widely used dataset in model editing, originally developed for zero-shot relation extraction. Following its adaptation by De Cao et al. (2021), the original answers were replaced with counterfactual ones to ensure that models had no prior exposure to the target facts, thereby providing a reliable benchmark for evaluating editing methods.

**COUNTERFACT** (Meng et al., 2022) is a challenging dataset specifically curated for model editing. It comprises 21,919 counterfactual statements whose target answers are initially assigned low probabilities by models, making it a rigorous benchmark for evaluating editing techniques on more challenging modifications.

**WikiBigEdit** (Thede et al., 2025) is a large-scale lifelong editing benchmark built through a fully automated data extraction pipeline that continuously incorporates new factual edits, ensuring long-term applicability for factuality evaluation. Its initial release contains over 500K question–answer pairs, providing a realistic setting for large-scale factual updates and better approximating deployment-time requirements.

Unlike the counterfactual knowledge in ZsRE and COUNTERFACT, WikiBigEdit consists of real-world facts from Wikipedia, some of which may have already been memorized by the evaluated LLMs. To ensure the learning of new knowledge, we excluded all samples that could be correctly answered by any of the three LLMs.

### A.1.4   REPRESENTATIVE TASKS

We adopt *Capability*, measured via the lm-evaluation-harness (Gao et al., 2024), in place of the traditional *Locality* metric (Yao et al., 2023) to more directly and rigorously assess whether editing perturbs unrelated knowledge and capabilities (Yang et al., 2024a;b; Gupta et al., 2024).

**MMLU** (Hendrycks et al., 2021) is a massive multitask benchmark consisting of multiple-choice questions across 57 subjects, spanning elementary mathematics, US history, computer science, law, and other areas that are important for people to learn. To reduce the prohibitive evaluation cost of the full benchmark, we randomly sample 500 questions from each subject, resulting in a balanced test set of 28,500 instances for our capability evaluation.

**Natural Questions** (Kwiatkowski et al., 2019) is an open-domain question answering benchmark, where queries are drawn from real anonymized search requests issued to Google, and answers are annotated from corresponding Wikipedia pages by human participators. We adopt its test set of 3610 question–answer pairs to evaluate the capability of target LLMs.

Table 6: Performance of LocFT-BF and baseline methods on the medical editing benchmark.

| Method | LLaMA3-8B | | Mistral-7B | | Qwen2.5-7B | |
|---|---|---|---|---|---|---|
| | Rel. | Cap. | Rel. | Cap. | Rel. | Cap. |
| Pre-edited | - | 57.26 | - | 44.82 | - | 58.52 |
| RLEdit | 58.92 | 58.34 | 21.18 | 30.36 | 60.43 | **61.43** |
| UltraEdit | 71.40 | **58.79** | 44.52 | **44.40** | 56.24 | 61.02 |
| LocFT-BF | **98.28** | 56.92 | **98.17** | 41.46 | **98.06** | 59.92 |

**SST2** (Socher et al., 2013) is a sentiment classification benchmark derived from the Stanford Sentiment Treebank. It consists of single sentences from movie reviews, each annotated with binary sentiment labels (positive or negative, with neutral cases removed). We utilize its standard test set to evaluate the sentiment classification capability of examined LLMs.

**WMT** (Bojar et al., 2016) is a benchmark series from the Workshop on Machine Translation, covering multiple years and language pairs. In our experiments, we adopt the WMT16 German–English (de–en) dataset to evaluate the translation capability of target LLMs.

**GSM8K** (Grade School Math 8K) (Cobbe et al., 2021) is a benchmark of 8500 high quality linguistically diverse grade school math word problems. It was created by OpenAI to support the task of question answering on basic mathematical problems that require multi-step reasoning. We use its test set of 1319 problems to evaluate the mathematical reasoning ability of LLMs.

## A.2 APPLYING LOCFT-BF TO MOE MODELS

To evaluate the generalizability of LocFT-BF to unseen datasets and heterogeneous architectures, we extend our evaluation to a Mixture-of-Experts (MoE) model, Qwen1.5-MoE-A2.7B (Team, 2024), using 1000 samples from the QAEdit dataset (Yang et al., 2025). Specifically, we employ the default locating strat-

Table 5: Performance on MoE model.

| Qwen1.5-MoE-A2.7B | Rel. | Gen. | Cap. |
|---|---|---|---|
| Pre-edited | - | - | 47.10 |
| LocFT-BF | 96.30 | 40.70 | 45.56 |

egy, directly editing the down-projection matrix of MLP in the mid-to-late layers, (i.e., `model.layers.20.mlp.shared_expert.down_proj`). The results in Table 5 demonstrate that LocFT-BF and its default locating strategy generalizes strongly to both new data and architectures, achieving effective and stable editing without disrupting the model's original capabilities. This cross-dataset and cross-architecture robustness highlights a substantial advantage of LocFT-BF over prior editing techniques, which typically exhibit high sensitivity to dataset and model variation.

## A.3 EVALUATION ON MEDICAL EDITING

To assess the effectiveness of editing methods in specialized domains, we evaluate LocFT-BF and the two most competitive baselines, RLEdit and UltraEdit on the medical knowledge editing benchmark (Chen et al., 2025a), which comprises 930 medical edits. We apply these edits across three LLMs using the same tuning locations employed in the main experiments. Given the absence of rephrased prompts in this benchmark, our evaluation focuses on reliability and capability. As shown in Table 6, LocFT-BF effectively injects specialized medical knowledge while preserving the models' general capabilities. Moreover, it significantly outperforms baseline methods, demonstrating that the superiority of LocFT-BF extends beyond general-domain factual edits.

## A.4 INTERPRETING GENERALIZATION GAP ON COUNTERFACT

The observed generalization drop on COUNTERFACT is a common issue for most editing techniques. The underlying reason lies in the dataset's unique evaluation protocol for generalization: instead of direct paraphrasing the edit prompt, COUNTERFACT constructs its generalization prompt by prepending irrelevant text to the edit prompt, as present in Figure 7. Such disruptive context makes generalization particularly challenging, especially for most methods that learn new knowl-

Table 7: Performance of LocFT-BF with data augmentation on the COUNTERFACT dataset.

| Method | LLaMA3-8B | | | Mistral-7B | | | Qwen2.5-7B | | |
|---|---|---|---|---|---|---|---|---|---|
| | Rel. | Gen. | Cap. | Rel. | Gen. | Cap. | Rel. | Gen. | Cap. |
| Pre-edited | - | - | 57.26 | - | - | 44.82 | - | - | 58.52 |
| MEMIT | 71.90 | 48.47 | 19.30 | 37.83 | 28.17 | 15.59 | 68.37 | 40.90 | 44.00 |
| AlphaEdit | 94.27 | 39.90 | 54.09 | 6.67 | 6.70 | 15.47 | 32.17 | 18.10 | 17.46 |
| LocFT-BF | 99.73 | 33.23 | 57.13 | 99.67 | 39.53 | 41.46 | 99.73 | 11.77 | 58.53 |
| **LocFT-BF-Aug** | 99.63 | **50.93** | 56.99 | 99.50 | **49.70** | 42.21 | 99.73 | **44.90** | 58.36 |

```
"Edit Prompt"      : "What sport does Dave Winfield play? They play",
"Rephrased Prompt" : "Andrey Tcheboharev\n( 9.) What sport does Dave Winfield play? They play",

"Augmented Context Templates":  ["The 2019-20 season has been. {}", "Therefore, we must not
↪  forget the importance of. {}", "Because I am a woman: The impact of. {}", "I have to admit,
↪  I was a bit. {}", "I have always been a fan of the. {}"]
```

Figure 7: An illustrative example from the COUNTERFACT dataset, including an edit prompt and its corresponding rephrased prompt. The figure also presents representative augmented context templates used by MEMIT and AlphaEdit, where the edit prompt is inserted into the placeholder {} to form diverse training prompts.

edge only from the edit prompt. However, methods such as MEMIT and AlphaEdit explicitly enhance robustness by augmenting each edit prompt with several randomly prefixed text, as shown in Figure 7, which explains their relative advantage on COUNTERFACT. Therefore, the lower generalization of LocFT-BF on this dataset reflects the unique evaluation design rather than the limitation of fine-tuning, and can be mitigated by incorporating prefix-based augmentation if necessary.

To substantiate this hypothesis, we apply LocFT-BF on the COUNTERFACT dataset across three LLMs and adopt the same data-augmentation strategy as MEMIT and AlphaEdit, augmenting each edit prompt with five randomly prefixed texts. As shown in Table 7, this simple augmentation markedly enhances generalization performance, placing LocFT-BF ahead of the augmented MEMIT and AlphaEdit, while preserving the general capabilities of edited models. These results further confirm that the perceived generalization gap stems from the evaluation protocol and can be effectively bridged via data augmentation.

## A.5 IMPLEMENTATION AND VALIDATION ON LLAMAFACTORY

The design simplicity of LocFT-BF, which relies solely on standard mini-batch training and localized parameter updates, facilitates seamless integration with existing fine-tuning frameworks. To demonstrate this, we incorporate LocFT-BF into the widely-used LlamaFactory library. This implementation not only validates the ease of reproduction of our method but also highlights its potential for efficient industrial deployment.

We employ the LlamaFactory implementation of LocFT-BF to replicate the experiments from Section 4, with comparative results shown in Table 8. Despite exhibiting marginal variance in reliability and capability, this implementation continues to significantly outperform existing editing baselines. Notably, it achieves substantial gains in generalization and efficiency, further extending the lead over competing methods. We attribute these variations to low-level implementation differences between the frameworks, such as the granularity of loss computation (token-level averaging in LlamaFactory and sample-level in ours) and the underlying engineering infrastructure. Collectively, these findings underscore the excellent adaptability and cross-framework robustness of LocFT-BF.

Furthermore, the comprehensive ecosystem of LlamaFactory facilitates a direct comparison with Full-Parameter Fine-Tuning (FullFT). As presented in Table 8, while FullFT achieves superior reliability and generalization on target edits, it incurs catastrophic degradation in the model's general

Table 8: Performance comparison of our original LocFT-BF against LlamaFactory (`LF`) implementations of both LocFT-BF and FullFT. The best results are denoted in **bold** and the second-best results are underlined.

| Method | LLaMA3-8B | | | | Mistral-7B | | | | Qwen2.5-7B | | | |
|---|---|---|---|---|---|---|---|---|---|---|---|---|
| | Rel. | Gen. | Cap. | Time | Rel. | Gen. | Cap. | Time | Rel. | Gen. | Cap. | Time |
| Pre-edited | – | – | 57.26 | – | – | – | 44.82 | – | – | – | 58.52 | – |
| **ZsRE** | | | | | | | | | | | | |
| FullFT (`LF`) | 98.47 | **87.57** | 40.44 | 0.21 | 96.43 | **87.40** | 16.07 | 0.20 | **99.17** | **84.73** | 59.05 | 0.20 |
| LocFT-BF (`LF`) | 95.07 | 79.10 | 56.03 | **0.04** | 95.00 | 71.43 | 41.38 | **0.04** | 98.77 | 75.67 | 60.83 | **0.07** |
| **LocFT-BF** | **98.97** | 75.83 | **57.04** | 0.27 | **98.93** | 49.57 | **42.73** | 0.31 | 98.87 | 74.37 | 59.07 | 0.49 |
| **COUNTERFACT** | | | | | | | | | | | | |
| FullFT (`LF`) | 99.63 | **95.07** | 31.53 | 0.22 | 99.53 | **95.67** | 15.64 | 0.21 | 99.70 | **76.23** | 53.53 | 0.20 |
| LocFT-BF (`LF`) | 99.07 | 71.53 | 56.58 | **0.04** | 98.77 | 58.60 | **41.70** | **0.04** | 99.43 | 41.80 | 59.27 | **0.05** |
| **LocFT-BF** | **99.73** | 33.23 | **57.13** | 0.38 | **99.67** | 39.53 | 41.46 | 0.24 | **99.73** | 11.77 | 58.53 | 0.48 |
| **WikiBigEdit** | | | | | | | | | | | | |
| FullFT (`LF`) | 99.43 | **96.57** | 46.69 | 0.21 | 98.73 | **95.33** | 18.23 | 0.21 | **99.50** | **96.17** | 58.08 | 0.21 |
| LocFT-BF (`LF`) | 97.43 | 88.20 | 55.51 | **0.03** | 95.20 | 84.23 | 42.13 | **0.04** | 98.33 | 82.67 | 59.03 | **0.08** |
| **LocFT-BF** | 98.87 | 73.77 | **56.93** | 0.54 | **98.90** | 74.97 | **42.39** | 0.39 | 99.43 | 53.30 | **59.83** | 0.98 |

Table 9: Scalability of baseline model editing methods on larger models.

| Method | Qwen2.5-7B | | | Qwen2.5-14B | | | Qwen2.5-14B | | |
|---|---|---|---|---|---|---|---|---|---|
| | Rel. | Gen. | Cap. | Rel. | Gen. | Cap. | Rel. | Gen. | Cap. |
| Strategy | | | | Default Position | | | Proportional Position | | |
| Pre-edited | - | - | 58.52 | - | - | 62.56 | - | - | 62.56 |
| RLEdit | 37.30 | 24.90 | 60.62 | 8.90 | 5.70 | 60.46 | 30.50 | 19.60 | 62.19 |
| UltraEdit | 44.10 | 19.00 | **60.91** | 12.50 | 8.10 | **62.97** | 34.20 | 22.30 | **63.07** |
| LocFT-BF | **99.20** | **83.40** | 59.33 | **99.00** | **78.80** | 61.62 | **99.50** | **77.90** | 61.12 |

capabilities. This severe overfitting confirms that despite its efficacy in memorizing specific samples, FullFT is too destructive to be a viable solution for practical, continuous model editing.

## A.6 SCALING BASELINE METHODS TO LARGER MODELS

Due to the inherent technical constraints hindering the parallel training of most model editing techniques, we select the two most competitive and scalable baselines, RLEdit and UltraEdit, and evaluate them on Qwen2.5-14B using 1000 ZsRE samples. As both methods require choosing editing layers, we employ the two layer-selection strategies used for LocFT-BF: ❶ Default Position: directly using the same absolute editing locations as in the 7B model. ❷ Proportional Position: scaling the locations proportionally to model depth. As presented in Table 9, both RLEdit and UltraEdit exhibit substantial performance degradation even at 14B, indicating that their practical scalability is severely limited, whereas LocFT-BF not only scales more easily from an engineering perspective but also maintains significantly better performance.

