# OpenReview forum: "Fine-tuning Done Right in Model Editing"
_ICLR.cc/2026/Conference — ICLR 2026 Poster_

### Official Review · Reviewer_Dcsj · 2025-10-20

**Soundness:** 3
**Presentation:** 3
**Contribution:** 3
**Rating:** 8
**Confidence:** 4

**Summary:**

This paper revisits the long-standing belief that fine-tuning is ineffective for model editing in LLMs. The authors argue that prior failures stem not from inherent limitations of fine-tuning but from mis-specified implementations that used a depth-first (DF), sample-wise pipeline rather than the standard breadth-first (BF), epoch-based, mini-batch paradigm. Through controlled experiments, they show that restoring the BF pipeline substantially improves reliability and preserves general capabilities.

Building on this insight, they introduce LocFT-BF (Localized Fine-Tuning with Breadth-First pipeline), which integrates three key ideas: (1) Restoring the standard BF pipeline; (2) Using mini-batch gradient aggregation; (3) Strategically selecting tuning locations (favoring later-layer MLP down-projections).

Extensive experiments demonstrate that LocFT-BF outperforms representative methods (e.g., WISE, RLEdit, UltraEdit, MEMIT, RECT, AlphaEdit) across multiple datasets (ZsRE, COUNTERFACT, WikiBigEdit) and models (LLaMA-3, Mistral, Qwen2.5). It achieves high reliability (>98%), strong generalization, efficient scalability (up to 100K sequential edits), and applicability to relatively very large models (up to 72B parameters).

Overall, this is a strong paper with clear contributions, high experimental quality, and potential to reshape the model editing field. Addressing the generalization limitations and exploring broader use cases would further enhance its impact.

**Strengths:**

- This paper challenges a widely held misconception in model editing and repositions fine-tuning as a strong, scalable baseline, and it’s a original proposal. Specifically, this paper identifies the flawed depth-first pipeline as the root cause of prior failures, an insight both simple and impactful. Furthermore, this paper proposes LocFT-BF, which combines principled pipeline design and layer selection with compelling empirical validation.
- The experimental design is thorough, e.g., controlled comparisons of DF vs BF, batch size studies, layer/module analyses, and large-scale evaluations (100K edits, 72B LLMs).
- Benchmarks against a broad set of representative baselines, including parameter-extension, meta-learning, and locate-then-edit methods. Results are consistently strong across multiple datasets and architectures, lending credibility to claims.
- The paper is well-structured, starting with the misconception, then systematically dissecting causes, followed by a constructive solution.
- Writing is clear and accessible even for readers not deeply familiar with model editing. Figures and tables (e.g., pipeline comparisons, tuning location studies, scaling results) are intuitive and reinforce key arguments.
- LocFT-BF is lightweight, plug-and-play, and avoids architectural overheads. It could shift the research landscape by elevating fine-tuning from a dismissed baseline to a leading paradigm for model editing. Furthermore, scalability to 100K edits and 72B models addresses real-world deployment needs, setting a new bar for evaluation standards.

**Weaknesses:**

- The generality of layer selection is not strong enough. The paper proposes later-layer MLP-down projections as relatively stable tuning locations, but acknowledges that optimal locations vary across models. More justification or theory behind this heuristic would strengthen the contribution.
- LocFT-BF underperforms on generalization for certain datasets (e.g., COUNTERFACT, where prompts involve irrelevant text). Authors suggest data augmentation as a fix in A.2, but no experiments directly test this claim.
- While diverse, the benchmark datasets are mostly factual-editing oriented. Broader evaluation (e.g., safety edits, temporal consistency) would strengthen claims about “real-world editing”.
- While comparisons are comprehensive within model editing literature, it would be useful to position LocFT-BF against emerging post-training alignment techniques (e.g., preference tuning, instruction backfitting) that could also serve as editing strategies.

**Questions:**

- How sensitive is LocFT-BF to tuning location selection in unseen LLM architectures (e.g., encoder-only, mixture-of-experts)? Could an automatic tuning-location search mechanism generalize better? How do you judge the layer selection robustness?
- For COUNTERFACT, LocFT-BF shows weaker generalization compared to some locate-then-edit baselines. Could integrating data augmentation or paraphrase generation directly into LocFT-BF resolve this gap? Could you provide some empirical study?
- Beyond factual edits, How would LocFT-BF perform on edits involving style adaptation, or safety constraints? Are there plans to extend evaluation beyond factual QA?
- While the method scales to 100K edits, does edit interference accumulate subtly over time (e.g., conflicting edits on related entities)? Have the authors analyzed error patterns at scale?
- How would LocFT-BF integrate into a real-world setting where edits arrive asynchronously over time and may be reverted? Is incremental updating compatible with LocFT-BF’s batch/epoch paradigm?

---

> ### Author Response · Authors · 2025-11-18
> **Response to Reviewer Dcsj (1/2)**
>
> Dear Reviewer Dcsj,
>
> We deeply appreciate your thorough review and supportive feedback. We are greatly encouraged by your recognition of our work, particularly its clear contributions, high experimental quality, and **potential to reshape the model editing field**.
>
> **W #1 and Q #1: generality of layer selection**
>
> While the exact optimal tuning locations may vary across architectures, our empirical study across three LLM families reveals a **consistent cross-model pattern**: later-layer MLP down-projection modules provide stable and high-quality edits. This pattern offers a **strong and practically reliable default strategy** for applying LocFT-BF to new models. Developing a full theoretical understanding of optimal layer selection is a broader and more fundamental question, which extends beyond the scope of this work and remains open even in related domains.
>
> To assess robustness on unseen architectures and datasets, we further evaluate LocFT-BF on a **MoE model**, `Qwen1.5-MoE-A2.7B`, using 1K samples from a **new dataset**, QAEdit [1]. Following our default strategy, we directly edit a later-layer MLP down-projection,  `model.layers.20.mlp.shared_expert.down_proj`. As shown in the table below, this default location **generalizes strongly to both unseen data and heterogeneous architectures**, achieving effective and stable editing without disrupting original capabilities.
>
> |                   | Reliability | Generalization | Capability |
> | ----------------- | ----------- | -------------- | ---------- |
> | Qwen1.5-MoE-A2.7B | -           | -              | 47.10      |
> | LocFT-BF          | 96.30       | 40.70          | 45.56      |
>
> [1] The Mirage of Model Editing: Revisiting Evaluation in the Wild. ACL 2025.
>
> **W #2 and Q #2: data augmentation for LocFT-BF on CounterFact dataset**
>
> Since the generalization gap on CounterFact mainly stems from its irrational prompt design and lies outside our paper's primary focus, we only provided an explanation in `Appendix A.2`. To directly address the reviewer's concern, we have now conducted an additional experiment verifying this claim.
>
> Specifically, we apply LocFT-BF to edit 3K CounterFact samples across three LLMs and adopt the same data-augmentation strategy as MEMIT and AlphaEdit, augmenting each edit prompt with five randomly prefixed texts. The results in the tables below **support our claim: this simple augmentation notably improves generalization, placing LocFT-BF above the augmented MEMIT and AlphaEdit**, while preserving the models' general capabilities.
>
> | LLaMA3-8B        | Reliability | Generalization | Capability |
> | ---------------- | ----------- | -------------- | ---------- |
> | MEMIT            | 71.90       | 48.47          | 19.30      |
> | AlphaEdit        | 94.27       | 39.90          | 54.09      |
> | LocFT-BF         | 99.73       | 33.23          | 57.13      |
> | **LocFT-BF-Aug** | 99.63       | **50.93**      | 56.99      |
>
> | Qwen2.5-7B       | Reliability | Generalization | Capability |
> | ---------------- | ----------- | -------------- | ---------- |
> | MEMIT            | 68.37       | 40.90          | 44.00      |
> | AlphaEdit        | 32.17       | 18.10          | 17.46      |
> | LocFT-BF         | 99.73       | 11.77          | 58.53      |
> | **LocFT-BF-Aug** | 99.73       | **44.90**      | 58.36      |
>
> | Mistral-7B       | Reliability | Generalization | Capability |
> | ---------------- | ----------- | -------------- | ---------- |
> | MEMIT            | 37.83       | 28.17          | 15.59      |
> | AlphaEdit        | 6.67        | 6.70           | 15.47      |
> | LocFT-BF         | 99.67       | 39.53          | 41.46      |
> | **LocFT-BF-Aug** | 99.50       | **49.70**      | 42.21      |

---

> ### Author Response · Authors · 2025-11-18
> **Response to Reviewer Dcsj (2/2)**
>
> **W #3 and Q #3: broader evaluation beyond factual QA**
>
> We appreciate the reviewer's suggestion to extend evaluation beyond factual QA. We would like to clarify that the primary goal of this work is to correct the long-standing misconception that fine-tuning is inherently ineffective for model editing. Therefore, we **focus on the standard, mainstream factual-knowledge editing benchmarks** that dominate prior evaluation protocols.
>
> To address the reviewer's concern, we evaluate LocFT-BF and two most competitive baseline methods on a **medical knowledge editing benchmark** [1], containing 930 medical edits. We apply these edits to three LLMs using the same tuning locations as in the paper. Since the benchmark does not provide rephrased prompts, we report reliability and capability only. As shown in the tables below, LocFT-BF **effectively injects medical knowledge while preserving the models' general capabilities**, and still significantly outperforms baseline methods, demonstrating that the method extends beyond general-domain factual edits.
>
> | LLaMA3-8B   | Original Model | RLEdit | UltraEdit | LocFT-BF |
> | ----------- | -------------- | ------ | --------- | -------- |
> | Reliability | -              | 58.92  | 71.40     | 98.28    |
> | Capability  | 57.26          | 58.34  | 58.79     | 56.92    |
>
> | Qwen2.5-7B  | Original Model | RLEdit | UltraEdit | LocFT-BF |
> | ----------- | -------------- | ------ | --------- | -------- |
> | Reliability | -              | 60.43  | 56.24     | 98.06    |
> | Capability  | 58.52          | 61.43  | 61.02     | 59.92    |
>
> | Mistral-7B  | Original Model | RLEdit | UltraEdit | LocFT-BF |
> | ----------- | -------------- | ------ | --------- | -------- |
> | Reliability | -              | 21.18  | 44.52     | 98.17    |
> | Capability  | 44.82          | 30.36  | 44.40     | 41.46    |
>
> [1] Beyond Memorization: A Rigorous Evaluation Framework for Medical Knowledge Editing. Arxiv 25.06.
>
> **W #4: comparison against post-training alignment techniques**
>
> We acknowledge the reviewer's insightful suggestion to compare model editing with alignment techniques. Although both occur after pretraining, they are designed for fundamentally different objectives and operate under different data regimes and evaluation criteria.
>
> Alignment techniques, such as preference tuning or instruction backfitting, aim to perform broad, distribution-level adjustments to a model's behavior, preferences, and style, using large, high-quality preference datasets.
>
> In contrast, model editing targets localized, fact-level modifications to the model's parametric knowledge, typically using small sets of QA or completion pairs. Besides, editing imposes strict constraints on preserving unrelated knowledge and capabilities, whereas alignment methods do not explicitly enforce such locality.
>
> Given these fundamental differences, alignment is not directly comparable to editing algorithms. Rather, the two techniques are complementary and can be combined in practice.
>
> **Q #4 : edit interference, conflicting edits, and error patterns at scale**
>
> Unlike the DF pipeline, which exhibits progressive performance degradation as edits accumulate, the BF pipeline maintains consistently high success rates (>98%) across scales without accumulation of interference.
>
> Regarding conflicting edits at scale, the BF pipeline inherently mitigates such issues: since all edits are optimized jointly, newly arriving edits can be checked, deduplicated, or reconciled **at the data level** before a new round of fine-tuning. This prevents contradictory edits from being injected and ensures that updates remain consistent even at large editing scales.
>
> We further examined error patterns using LLMs (e.g., GPT-5 and Claude-4.5) and manual inspection. The errors span a broad range of topics, including biography, history, geography, arts, sports, and science, mirroring the overall distribution of the dataset, and appear largely random rather than concentrated around any specific pattern.
>
> **Q #5: asynchronous edits over time**
>
> In real-world settings where edits arrive asynchronously, LocFT-BF can simply **integrate new edits incrementally** and perform a new round of fine-tuning. As shown in the efficiency analysis of `Section 4.2`, the fine-tuning step of LocFT-BF is sufficiently fast to make such incremental updates practical in deployment. Moreover, in practical applications, new edits can be accumulated into small batches before triggering the next update, which further reduces the computational overhead.
>
> We hope these clarifications have fully addressed your concerns. If you have any further suggestions, please feel free to reach out.

---

> ### Comment · Reviewer_Dcsj · 2025-11-22
> **Response to Rebuttal**
>
> Dear Authors,
>
> Thank you very much for the clarification! Most of them have addressed my concerns.
>
> I appreciate the response, especially the newly conducted experiments. Including those in the revised draft will strengthen the paper.
>
> Overall, I acknowledge the quality of this work, and will maintain my positive score.
>
> Best Regards,
>
> Reviewer Dcsj

---

> > ### Author Response · Authors · 2025-11-24
> > **Thank you for your encouraging response.**
> >
> > Dear Reviewer Dcsj,
> >
> > Thank you very much for your kind follow-up and for acknowledging the quality of our work. We are glad to hear that our responses have addressed your concerns. We will incorporate your valuable suggestions and further improve the manuscript in the revision.
> >
> > Once again, we sincerely appreciate your time, efforts, and supportive evaluation of our paper.
> >
> > Best regards,
> >
> > Authors

---

### Official Review · Reviewer_MVzX · 2025-10-23

**Soundness:** 3
**Presentation:** 3
**Contribution:** 3
**Rating:** 8
**Confidence:** 3

**Summary:**

This paper challenges the prevailing view that fine-tuning is a weak baseline for model editing. The authors compellingly argue that its reported failures stem not from the method itself, but from a flawed "Depth-First" (DF) implementation that processes edits sequentially and individually. This DF pipeline, they demonstrate, leads to catastrophic forgetting and degrades general capabilities.

The core proposal is to revert to the standard "Breadth-First" (BF) training pipeline, which uses epoch-based, mini-batch optimization over the entire set of edits. This is combined with a systematic study that identifies tuning later-layer MLP modules as an effective strategy for localizing updates.

The resulting method, LocFT-BF, is simple and highly effective. Extensive experiments show that it significantly outperforms state-of-the-art methods and, for the first time, successfully scales to 100,000 sequential edits and 72B-parameter models, setting a new and powerful baseline for the field.

**Strengths:**

Strength1: The claims are exceptionally well-supported by a series of controlled experiments that systematically isolate and analyze the impacts of the training pipeline (DF vs. BF), batch size, and tuning location.

Strength2: LocFT-BF is conceptually simple, efficient, and easy to implement, especially when compared to more complex editing techniques. This makes it a highly practical and powerful new baseline.

**Weaknesses:**

Weakness1: The method shows a generalization gap on the COUNTERFACT dataset compared to baselines that use data augmentation. While the authors provide a plausible explanation, the claim that this can be "readily mitigated" with augmentation is not experimentally verified in the paper.

**Questions:**

Q1: Have you performed any experiments to support the claim that the generalization gap on COUNTERFACT can be closed by applying data augmentation techniques similar to those used by methods like MEMIT or AlphaEdit?

Q2: How would a standard LoRA, when applied to the empirically-found optimal modules (e.g., later-layer MLPs) and trained using the correct BF pipeline, compare to your full-parameter local tuning approach in LocFT-BF? This would help clarify whether, under the proper training paradigm, the concept of localization is the more critical factor, or if full-parameter tuning is inherently superior to low-rank adaptation in editing.

---

> ### Author Response · Authors · 2025-11-18
> **Response to Reviewer MVzX**
>
> Dear Reviewer MVzX,
>
> We sincerely appreciate your constructive review and supportive score. We are encouraged by your recognition that **our claims are exceptionally well-supported**, and that LocFT-BF is conceptually simple, efficient, and easy to implement, **setting a new and powerful baseline for model editing field**.
>
> **W #1 and Q #1: data augmentation for LocFT-BF on CounterFact dataset**
>
> We acknowledge the reviewer's insightful suggestion to include empirical validation of our claim that the generalization gap on CounterFact can be mitigated through data augmentation. Since this gap mainly stems from the dataset's irrational prompt design and lies outside our paper's primary focus, we only provided an explanation in `Appendix A.2`. To directly address the reviewer's concern, we have now conducted an additional experiment verifying this claim.
>
> Specifically, we apply LocFT-BF to edit 3K samples from CounterFact across three LLMs and adopt the same data-augmentation strategy as MEMIT and AlphaEdit, augmenting each edit prompt with five randomly prefixed texts. As shown in the tables below, **this simple augmentation markedly enhances generalization performance, placing LocFT-BF ahead of the augmented MEMIT and AlphaEdit**, while preserving the general capabilities of edited models. These results **directly support our claim** that the generalization gap can be readily mitigated through data augmentation.
>
> | LLaMA3-8B        | Reliability | Generalization | Capability |
> | ---------------- | ----------- | -------------- | ---------- |
> | MEMIT            | 71.90       | 48.47          | 19.30      |
> | AlphaEdit        | 94.27       | 39.90          | 54.09      |
> | LocFT-BF         | 99.73       | 33.23          | 57.13      |
> | **LocFT-BF-Aug** | 99.63       | **50.93**      | 56.99      |
>
> | Qwen2.5-7B       | Reliability | Generalization | Capability |
> | ---------------- | ----------- | -------------- | ---------- |
> | MEMIT            | 68.37       | 40.90          | 44.00      |
> | AlphaEdit        | 32.17       | 18.10          | 17.46      |
> | LocFT-BF         | 99.73       | 11.77          | 58.53      |
> | **LocFT-BF-Aug** | 99.73       | **44.90**      | 58.36      |
>
> | Mistral-7B       | Reliability | Generalization | Capability |
> | ---------------- | ----------- | -------------- | ---------- |
> | MEMIT            | 37.83       | 28.17          | 15.59      |
> | AlphaEdit        | 6.67        | 6.70           | 15.47      |
> | LocFT-BF         | 99.67       | 39.53          | 41.46      |
> | **LocFT-BF-Aug** | 99.50       | **49.70**      | 42.21      |
>
> **Q #2: apply LoRA to empirically-found optimal modules using BF pipeline**
>
> We thank the reviewer for raising this insightful question that we have also examined in our study but omitted the results from the paper due to their consistently poor performance. Specifically, we applied standard LoRA with different ranks to the empirically identified optimal modules (e.g., `layer22.mlp.down_proj` of LLaMA3-8B) under the correct BF pipeline. The results in the following tables demonstrate that **applying LoRA to localized modules exhibits extremely low editing success**.
>
> A plausible explanation is that combining low-rank adaptation with localized module updates **restricts the number of trainable parameters to less than 0.005%** of the model, which may constrain the learning capacity for new knowledge. These findings suggest that localization is crucial to the success of LocFT-BF, yet sufficient parameter expressiveness within those localized regions is equally important.
>
> | rank=8,lora_alpha=16 | Reliability | Generalization | Capability |
> | -------------------- | ----------- | -------------- | ---------- |
> | LLaMA3-8B            | 1.40        | 1.47           | 55.93      |
> | Qwen2.5-7B           | 3.83        | 3.43           | 60.64      |
> | Mistral-7B           | 5.67        | 5.10           | 43.77      |
>
> | rank=32,lora_alpha=64 | Reliability | Generalization | Capability |
> | --------------------- | ----------- | -------------- | ---------- |
> | LLaMA3-8B             | 6.23        | 4.47           | 56.94      |
> | Qwen2.5-7B            | 5.77        | 4.53           | 60.25      |
> | Mistral-7B            | 7.97        | 6.67           | 42.25      |
>
> We will incorporate the supplementary experiments into the revised manuscript and continue to refine our work. We hope these clarifications have fully addressed your concerns. If you have any further questions, please feel free to reach out.

---

> > ### Comment · Reviewer_MVzX · 2025-11-25
> >
> > Dear Authors,
> >
> > Thanks for your detailed response and for conducting the requested experiments. I appreciate how this work clarifies the role of fine-tuning in model editing.
> >
> > I will maintain my positive score and have increased my confidence rating.
> >
> > Best,
> >
> > Reviewer MVzX

---

> > > ### Author Response · Authors · 2025-11-25
> > > **Thank you for your kind feedback.**
> > >
> > > Dear Reviewer MVzX,
> > >
> > > Thank you very much for your kind feedback and for your acknowledgment of our work. We are glad to hear that our responses have addressed your concerns. We will incorporate your valuable suggestions to further refine the manuscript in the revision.
> > >
> > > Once again, we deeply appreciate the time, effort, and expertise you have dedicated to reviewing our work!
> > >
> > > Best regards,
> > >
> > > Authors

---

### Official Review · Reviewer_EfTQ · 2025-10-27

**Soundness:** 2
**Presentation:** 3
**Contribution:** 2
**Rating:** 4
**Confidence:** 4

**Summary:**

This paper focuses on the long-term underestimation of fine-tuning methods in the field of large language model (LLMs) model editing. Through in-depth analysis of the shortcomings of existing fine-tuning implementations, an innovative LocFT-BF method is proposed, which restores fine-tuning to a standard breadth first (BF) pipeline and combines local parameter tuning. The paper verified the effectiveness of the method through systematic control experiments, surpassing existing mainstream model editing methods on multiple models and datasets, and achieving stable editing of 100000 edits and 72B parameter models for the first time. The research design is rigorous, the experimental scale is sufficient, and it has important theoretical and practical significance for the field of model editing. However, there is still room for improvement in terms of theoretical depth, adaptability to special scenarios, and supplementary experimental details in the paper.

**Strengths:**

- Overturns the long-standing misconception that fine-tuning is unsuitable for model editing , proving through rigorous experiments that prior failures were due to flawed implementations (i.e., the depth-first pipeline) rather than inherent limitations.
- The proposed LocFT-BF method is exceptionally simple and efficient, requiring no complex architectural changes, auxiliary data, or costly pre-computation overhead typical of other editing techniques。
- Demonstrates unprecedented scalability by showing LocFT-BF effectively handles 100K sequential edits and scales to 72B-parameter models, addressing a key practical bottleneck in model editing.
- Provides a systematic and in-depth empirical analysis of how to apply fine-tuning correctly, including rigorous studies on the training pipeline , batch size , and optimal parameter tuning locations.
- Achieves state-of-the-art performance, comprehensively and significantly outperforming existing methods, particularly in challenging large-scale lifelong editing scenarios.

**Weaknesses:**

- The paper's justification for the superiority of the Breadth-First (BF) pipeline relies heavily on empirical phenomena (e.g., mitigating overwriting) . It lacks a deeper theoretical analysis, such as convergence or stability bounds, to formally explain why mini-batch optimization provides superior stability and forgetting mitigation compared to the sample-wise, depth-first (DF) approach in the context of sequential editing.
- The "Capability" evaluation, while broad, is confined to standard benchmarks (e.g., MMLU, GSM8K, SST2) that are predominantly multi-choice or classification tasks. This setup overlooks more complex and valuable scenarios, failing to test the model's robustness against, for example, generating hallucinations (e.g., TruthfulQA) or handling complex reasoning (e.g., BBH) post-edit.
- The paper repeatedly claims that the observed degradation in "Generalization" (e.g., on COUNTERFACT) can be mitigated by data augmentation . However, this claim remains unsubstantiated, as no supporting experiments are provided in the main text to validate whether augmentation can indeed recover generalization without negatively impacting reliability or capability. The author can refer to the following work on augmentation and discuss it:

[1] Allen-Zhu Z, Li Y. Physics of language models: Part 3.1, knowledge storage and extraction[J]. arXiv preprint arXiv:2309.14316, 2023.

[2] Jiang K, Du Y, Ding Y, et al. When Large Multimodal Models Confront Evolving Knowledge: Challenges and Pathways[J]. arXiv preprint arXiv:2505.24449, 2025.

[3] Wang K, Zhu J, Ren M, et al. A survey on data synthesis and augmentation for large language models[J]. arXiv preprint arXiv:2410.12896, 2024.

[4] Park C F, Zhang Z, Tanaka H. $\textit {New News} $: System-2 Fine-tuning for Robust Integration of New Knowledge[J]. arXiv preprint arXiv:2505.01812, 2025.

- The experimental validation focuses exclusively on general-domain factual knowledge (e.g., ZsRE, COUNTERFACT). The effectiveness and generalizability of LocFT-BF in editing specialized, high-stakes domains (e.g., biological, financial, or legal knowledge) or in cross-domain/cross-lingual editing scenarios remain unverified.
- The analysis of scaling to larger models (§5.2)  only evaluates LocFT-BF on the Qwen2.5 series. It crucially lacks a direct comparison against baseline methods (e.g., RLEdit, UltraEdit) at these larger scales (14B, 32B, 72B). This omission makes it difficult to substantiate the strong claim that LocFT-BF is uniquely or more easily scalable than other methods while maintaining performance.

If the author can solve the above problems, I will consider increasing the score.

**Questions:**

- Beyond the empirical evidence of catastrophic overwriting in the DF pipeline , what is the theoretical justification for why the BF pipeline with mini-batching  provides superior stability and mitigates catastrophic forgetting so effectively during sequential editing?
- How does LocFT-BF perform on benchmarks designed to test for hallucination (e.g., TruthfulQA) or complex multi-step reasoning (e.g., BBH)? Does the localized fine-tuning approach maintain robustness in these more challenging generative scenarios post-editing?
- The authors argue that the generalization gap can be offset by data augmentation . Have experiments been conducted to support this claim, and if so, does augmentation successfully improve generalization without compromising the method's high reliability and capability scores?
- How well does the LocFT-BF framework, including its specific layer-selection strategy (e.g., later-layer MLPs) , generalize to editing highly specialized, high-stakes knowledge domains (e.g., legal or medical) or to multilingual knowledge editing tasks? Although this article proposes a new editing method, it focuses more on the analysis of knowledge editing methods. Therefore, we hope that the author can explore more extensively and provide more valuable insights.

[1] Zhang X, Liang Y, Meng F, et al. Multilingual knowledge editing with language-agnostic factual neurons[J]. arXiv preprint arXiv:2406.16416, 2024.

[2] Wang W, Haddow B, Birch A. Retrieval-augmented multilingual knowledge editing[J]. arXiv preprint arXiv:2312.13040, 2023.

[3] Wei Z, Deng J, Pang L, et al. Mlake: Multilingual knowledge editing benchmark for large language models[J]. arXiv preprint arXiv:2404.04990, 2024.

[4] Rosati D, Gonzales R, Chen J, et al. Long-form evaluation of model editing[J]. arXiv preprint arXiv:2402.09394, 2024.

- While LocFT-BF scales to 72B models, this analysis lacks a baseline comparison. Could a comparison be provided, even on a mid-sized model (e.g., 32B), to better substantiate the claim of superior scalability, or is the assumption that other methods are computationally infeasible at this scale verified?

---

> ### Author Response · Authors · 2025-11-18
> **Response to Reviewer EfTQ (1/3)**
>
> Dear Reviewer EfTQ,
>
> We sincerely appreciate your thorough review and constructive feedback. We are grateful that you recognize the strengths of our work, particularly in **overturning the long-standing misconception about fine-tuning in model editing** and providing systematic and rigorous analysis.
>
> **W #1 and Q #1: theoretical analysis for the superiority of BF pipeline**
>
> The core contribution of our work is to reveal that **prior fine-tuning based editing methods adopt a non-standard, sample-wise DF pipeline**, and that correcting this to a standard mini-batch BF pipeline restores superior effectiveness of fine-tuning for model editing. Our goal is **not to re-prove the theoretical foundations of mini-batch optimization**, which has been well understood for decades.
>
> Here, we provide a brief theoretical analysis on why mini-batch optimization offers improved stability and mitigates overwriting.
>
> (1) Impact of training pipeline.
>
> Consider a sequence of editing samples $S = \lbrace s_1, s_2, \ldots, s_N \rbrace$,  where each sample $s_i = (x_i, y_i)$ is a question–answer pair, and let $\theta$ denotes the model parameters.
>
> In the DF pipeline, at time step $t$, the model is exposed only to the current sample $s_t$. We can formalize this with a **sequential conditional loss**:
>
> $$
> \mathcal{L}\_{DF}(\theta; t) = \ell(f\_\theta(x\_t), y\_t) \quad \text{for time } t
> $$
>
> The optimization at step $t$ is given by: $\theta\_t = \arg\min\_{\theta} \mathcal{L}\_{DF}(\theta; t)$. From the perspective of the **stability–optimization trade-off**: since $\mathcal{L}_{DF}$ contains **no constraints from earlier samples**, the model aggressively minimizes a single-sample objective, making overwriting of previous knowledge almost inevitable.
>
> In contrast, the BF pipeline optimizes the **joint loss** over all editing samples:
>
> $$
> \mathcal{L}\_{BF}(\theta) = \mathbb{E}\_{(x,y)\sim S}\big[\ell(f\_\theta(x), y)\big]
>      \approx \frac{1}{N}\sum\_{i=1}^{N} \ell(f\_\theta(x\_i), y\_i)
> $$
>
> This objective ensures that all edited knowledge is **simultaneously represented** in the parameter space, naturally mitigating catastrophic overwriting.
>
> (2) Impact of gradient aggregation.
>
> For the DF pipeline, SGD update rule is $\theta_{t+1} = \theta_t - \eta_t g_t, \text{ where } g_t = \nabla \ell(f_\theta(x_t), y_t)$ is the gradient computed on a single example. Although $g_t$ is an unbiased estimator of the true gradient $\nabla \mathcal{L}(\theta_t)$, its variance $\mathbb{E}[||g_t - \nabla \mathcal{L}(\theta_t)||^2]$ can be large in practice.
>
> Under a mini-batch of size $B$, the gradient estimate becomes $\hat{g}\_t = \frac{1}{B} \sum\_{i=1}^B \nabla \ell(f\_\theta(x\_i), y\_i)$, whose variance satisfies approximately $\text{Var}(\hat{g}_t) \approx \frac{1}{B} \text{Var}(g_t)$, **yielding a substantially more stable estimate of the true gradient and more stable parameter updates**.

---

> ### Author Response · Authors · 2025-11-18
> **Response to Reviewer EfTQ (2/3)**
>
> **W #2 and Q #2: the breadth of Capability evaluation**
>
> We would like to clarify a potential misunderstanding: our Capability evaluation is **not confined to multiple-choice or classification tasks**. As described in `Section 4.1` and `Appendix A.1.4`, we evaluate five diverse task types: classification (SST2), multiple-choice (MMLU), question answering (Natural Questions), translation (WMT), and mathematical reasoning (GSM8K). The suggested TruthfulQA and BBH are **aligned in task nature** with Natural Questions and GSM8K.
>
> To directly address the reviewer's concern, we apply LocFT-BF to three LLMs using 3K CounterFact edits, and **assess the edited models on TruthfulQA**. As shown in the table below, LocFT-BF preserves performance on TruthfulQA, indicating that localized fine-tuning maintains robustness even in hallucination-sensitive task.
>
> | TruthfulQA     | LLaMA3-8B | Qwen2.5-7B | Mistral-7B |
> | -------------- | --------- | ---------- | ---------- |
> | Original Model | 36.23     | 47.98      | 28.03      |
> | LocFT-BF       | 35.74     | 46.27      | 28.89      |
>
> **W #3 and Q #3: data augmentation for LocFT-BF on CounterFact dataset**
>
> We appreciate the reviewer's insightful suggestion and the relevant references. Since the generalization gap on CounterFact mainly **stems from its irrational prompt design** and lies outside our paper's primary focus, we only provided an analytical explanation in `Appendix A.2`. To directly address the reviewer's concern, we have now conducted an additional experiment to validate this claim.
>
> Specifically, we apply LocFT-BF to edit 3K CounterFact samples across three LLMs and adopt the same data-augmentation strategy as MEMIT and AlphaEdit, augmenting each edit prompt with five randomly prefixed texts. The results in the tables below **support our claim: this simple augmentation notably improves generalization, placing LocFT-BF above the augmented MEMIT and AlphaEdit**, while preserving the models' general capabilities.
>
> | LLaMA3-8B        | Reliability | Generalization | Capability |
> | ---------------- | ----------- | -------------- | ---------- |
> | MEMIT            | 71.90       | 48.47          | 19.30      |
> | AlphaEdit        | 94.27       | 39.90          | 54.09      |
> | LocFT-BF         | 99.73       | 33.23          | 57.13      |
> | **LocFT-BF-Aug** | 99.63       | **50.93**      | 56.99      |
>
> | Qwen2.5-7B       | Reliability | Generalization | Capability |
> | ---------------- | ----------- | -------------- | ---------- |
> | MEMIT            | 68.37       | 40.90          | 44.00      |
> | AlphaEdit        | 32.17       | 18.10          | 17.46      |
> | LocFT-BF         | 99.73       | 11.77          | 58.53      |
> | **LocFT-BF-Aug** | 99.73       | **44.90**      | 58.36      |
>
> | Mistral-7B       | Reliability | Generalization | Capability |
> | ---------------- | ----------- | -------------- | ---------- |
> | MEMIT            | 37.83       | 28.17          | 15.59      |
> | AlphaEdit        | 6.67        | 6.70           | 15.47      |
> | LocFT-BF         | 99.67       | 39.53          | 41.46      |
> | **LocFT-BF-Aug** | 99.50       | **49.70**      | 42.21      |

---

> ### Author Response · Authors · 2025-11-18
> **Response to Reviewer EfTQ (3/3)**
>
> **W #4 and Q #4: evaluation on specialized domains**
>
> It is important to clarify that the primary goal of this work is to correct the long-standing misconception that fine-tuning is inherently ineffective for model editing. Accordingly, we **focus on the standard, mainstream factual-knowledge editing benchmarks** that dominate prior evaluation protocols, rather than exhaustively evaluate across all possible domains
>
> To address the reviewer's curiosity about specialized domains, we additionally evaluate LocFT-BF and two most competitive baseline methods on a **medical knowledge editing benchmark** [1], containing 930 medical edits. We apply these edits to three LLMs using the same tuning locations as in the paper. Since the benchmark does not provide rephrased prompts, we report reliability and capability only. As shown in the tables below, LocFT-BF **effectively injects medical knowledge while preserving the models' general capabilities**, and still significantly outperforms baseline methods, demonstrating that the method extends beyond general-domain factual edits.
>
> | LLaMA3-8B   | Original Model | RLEdit | UltraEdit | LocFT-BF |
> | ----------- | -------------- | ------ | --------- | -------- |
> | Reliability | -              | 58.92  | 71.40     | 98.28    |
> | Capability  | 57.26          | 58.34  | 58.79     | 56.92    |
>
> | Qwen2.5-7B  | Original Model | RLEdit | UltraEdit | LocFT-BF |
> | ----------- | -------------- | ------ | --------- | -------- |
> | Reliability | -              | 60.43  | 56.24     | 98.06    |
> | Capability  | 58.52          | 61.43  | 61.02     | 59.92    |
>
> | Mistral-7B  | Original Model | RLEdit | UltraEdit | LocFT-BF |
> | ----------- | -------------- | ------ | --------- | -------- |
> | Reliability | -              | 21.18  | 44.52     | 98.17    |
> | Capability  | 44.82          | 30.36  | 44.40     | 41.46    |
>
> [1] Beyond Memorization: A Rigorous Evaluation Framework for Medical Knowledge Editing. Arxiv 25.06.
>
> **W #5 and Q #5: evaluating baseline methods on larger models**
>
> The scalability advantage of LocFT-BF stems not just from its superior performance on larger models, but **more importantly from its algorithmic simplicity and engineering compatibility**, which make it straightforward to scale to larger models. In contrast, existing methods are limited by the **complex and specialized algorithms**, which are computationally intensive, **poorly suited for parallelization**.
>
> To address the reviewer's concern, we attempted to apply RLEdit and UltraEdit, two most competitive and scalable baselines, on Qwen2.5-14B using 1K ZsRE samples. Since these methods also require selecting editing layers, we employed the two strategies used in LocFT-BF: (i) directly using the **same editing locations** as in the 7B model, and (ii) scaling the locations **proportionally** to model depth. As shown in the tables below, **both methods exhibit substantial performance degradation even at 14B**, indicating that their scalability is limited in practice, whereas LocFT-BF not only scales more easily from an engineering perspective but also maintains significantly better performance.
>
> | **Qwen2.5-7B** | Reliability | Generalization | Capability |
> | -------------- | ----------- | -------------- | ---------- |
> | Original Model | -           | -              | 58.52      |
> | RLEdit         | 37.30       | 24.90          | 60.62      |
> | UltraEdit      | 44.10       | 19.00          | 60.91      |
> | LocFT-BF       | 99.20       | 83.40          | 59.33      |
>
> | Same Locations  | Reliability | Generalization | Capability |
> | --------------- | ----------- | -------------- | ---------- |
> | **Qwen2.5-14B** | -           | -              | 62.56      |
> | RLEdit          | 8.90        | 5.70           | 60.46      |
> | UltraEdit       | 12.50       | 8.10           | 62.97      |
> | LocFT-BF        | 99.00       | 78.80          | 61.62      |
>
> | Proportional Locations | Reliability | Generalization | Capability |
> | ---------------------- | ----------- | -------------- | ---------- |
> | **Qwen2.5-14B**        | -           | -              | 62.56      |
> | RLEdit                 | 30.50       | 19.60          | 62.19      |
> | UltraEdit              | 34.20       | 22.30          | 63.07      |
> | LocFT-BF               | 99.50       | 77.90          | 61.12      |
>
> We hope these clarifications have fully addressed your concerns. If so, we would deeply appreciate your consideration in raising your score. If you have any further suggestions, please feel free to reach out.

---

> > ### Comment · Reviewer_EfTQ · 2025-11-19
> > **Response to Authors of Paper 926**
> >
> > Thank you for adopting my review comments and providing detailed responses. I have raised my score to 6 and wish you all the best.

---

> > > ### Author Response · Authors · 2025-11-19
> > > **Thank you for the supportive feedback.**
> > >
> > > Dear Reviewer EfTQ,
> > >
> > > Thank you very much for your kind follow-up. We are glad to hear that our responses have addressed your concerns. We sincerely appreciate your thoughtful feedback, and we will incorporate your valuable suggestions into the revised manuscript to further enhance its quality.
> > >
> > > If there are any remaining aspects you would like to discuss, we would be more than happy to engage further. Thank you again for your time and efforts on our paper.
> > >
> > > Best regards,
> > >
> > > Authors

---

### Official Review · Reviewer_QAtd · 2025-10-29

**Soundness:** 2
**Presentation:** 2
**Contribution:** 2
**Rating:** 4
**Confidence:** 4

**Summary:**

This paper addresses the failure of fine-tuning in model editing by introducing two principled improvements. The authors argue that the failure of existing fine-tuning approaches mainly stems from the depth-first (single-pass) updating paradigm, which leads to catastrophic forgetting. To mitigate this, they restore the standard breadth-first training pipeline, optimizing all mini-batches across epochs rather than editing samples sequentially. Furthermore, by systematically analyzing which layers and modules are most effective for editing, the study finds that tuning the down- or up-projection matrices in later layers generally achieves near-perfect editing success while maintaining general capabilities.

**Strengths:**

1. The paper conducts extensive experiments to validate their hypotheses and insights.

2. The authors’ observation that the depth-first approach leads to instability and catastrophic forgetting is indeed convincing in my view.

3. The paper provides clear and informative visualizations and ablation studies

**Weaknesses:**

1. While the breadth-first pipeline mitigates forgetting effectively, its computational cost grows linearly with the number of accumulated edits, as previous samples must be revisited in each fine-tuning pass. This may raise concerns about scalability in real-time or large-scale editing scenarios, despite the authors’ empirical demonstration of efficiency at 100K edits.

2. While the breadth-first method alleviates catastrophic forgetting to some extent, it does not fundamentally eliminate the issue, since there is no guarantee that the model’s pretrained knowledge remains intact after editing.

3. The locate strategy proposed in this paper appears somewhat vague and even inefficient. For example, in Section 5.2, the authors directly scale the editing location from Qwen2.5-14B to Qwen2.5-32B, which seems to rely on the assumption that the editing samples remain identical and that the two models share highly similar architectures. However, in real-world applications, the editing samples are dynamically updated, and the optimal editing module may vary accordingly. Simply transferring the previous settings to larger or more heterogeneous LLMs could therefore be unreliable, and the effectiveness of such an extension may degrade even further in those scenarios.

4. At the same time, the method this paper uses to identify the optimal tuning location also appears somewhat inefficient, as they fine-tune across all layers and five candidate modules in order to determine the best position.

**Questions:**

1. When new samples arrive, how does LocFT-BF dynamically reconstruct its epochs and batches? Moreover, how does it handle edits that conflict with previous ones?

2. For a new, heterogeneous LLM and an unseen task, could the authors clarify how LocFT-BF determines or generalizes the choice of the optimal editing module?

3. Could the authors provide a comparison of the time and computational cost between the breadth-first and depth-first fine-tuning pipelines?

If I have misunderstood anything, please feel free to point it out.

---

> ### Author Response · Authors · 2025-11-18
> **Response to Reviewer QAtd (1/2)**
>
> Dear Reviewer QAtd,
>
> We sincerely appreciate your time, effort, and thoughtful review of our work. We are delighted by your recognition of our extensive experiments, convincing analysis, and clear and informative visualizations and ablations.
>
> **W #1: efficiency of LocFT-BF in practical scenarios**
>
> As shown in the efficiency analysis of `Section 4.2`, LocFT-BF is **already highly efficient** in the field of model editing, achieving an average editing time less than 0.5 second per sample on a single GPU, which is over 50× faster than representative locate-then-edit methods such as AlphaEdit.
>
> Furthermore, since LocFT-BF is a standard supervised fine-tuning procedure, it can **directly benefit from the full ecosystem of fine-tuning optimizations**, including modern acceleration techniques and training frameworks. For example, after **implementing LocFT-BF in LLaMA-Factory**, the average editing time accelerates to 0.1 second per sample, which we will report in the appendix of the revised manuscript.
>
> At the same time, our experiments operate at 100K edits, which is over 30× larger than the typical scales evaluated in prior work (up to 3K edits) [1, 2]. This setting also **aligns with realistic deployment scenarios**, where editing is used as a lightweight knowledge patch between pretrained versions, rather than to perform large-scale knowledge acquisition as in pretraining. Under this practical scale, LocFT-BF remains highly efficient. Hence, **the scalability concern does not arise in practical deployments**.
>
> [1] AlphaEdit: Null-Space Constrained Knowledge Editing for Language Models. ICLR 2025.
>
> [2] WISE: Rethinking the Knowledge Memory for Lifelong Model Editing of Large Language Models. NIPS 2024.
>
> **W #2: elimination of catastrophic forgetting**
>
> It is important to emphasize that our goal is not to eliminate catastrophic forgetting in the sense pursued by continual learning research. Instead, our aim is to show that switching from DF to BF training pipeline substantially reduces **overwriting** between edits, thereby **repositioning fine-tuning as a superior technique** for model editing and **correcting the prevailing misconception** that FT is unsuitable for this setting.
>
> To assess the preservation of pretrained knowledge and capabilities, we evaluate the edited models on five representative downstream tasks: classification (SST-2), multiple-choice (MMLU), question answering (NQ), translation (WMT), and mathematical reasoning (GSM8K). The results show that **LocFT-BF effectively avoids catastrophic forgetting on pretrained knowledge and capabilities**, an outcome that full-parameter fine-tuning typically struggles to achieve.
>
> **W #3 and Q #2: locating strategy for new data and heterogeneous LLMs**
>
> From the data perspective, the identified tuning location is highly robust across datasets. As shown in `Section 4.2`, the location determined using 1K samples from ZsRE **remains stable and performs strongly on 3K samples from both CounterFact and WikiBigEdit**.
>
> From the model perspective, our extensive exploration across three different LLM families shows that, while the exact optimal module varies across models, there exists a clear cross-model pattern: **mid-to-late MLP down-projection matrices consistently yield superior editing performance**. This pattern provides a reliable and architecture-agnostic default locating strategy.
>
> To directly address the reviewer's concern, we further evaluate LocFT-BF on a **MoE model**, `Qwen1.5-MoE-A2.7B`, using 1K samples from a **new dataset**, QAEdit [1]. Specifically, we **directly apply the default strategy**, editing the mid-to-late MLP down-projection matrix, `model.layers.20.mlp.shared_expert.down_proj`. As shown in the table below, this **default location generalizes strongly to both unseen data and heterogeneous architectures**, achieving effective and stable editing without disrupting original capabilities.
>
> This cross-dataset and cross-architecture robustness also represents a substantial advantage over prior editing methods, which typically exhibit greater sensitivity to dataset and model variation.
>
> |                   | Reliability | Generalization | Capability |
> | ----------------- | ----------- | -------------- | ---------- |
> | Qwen1.5-MoE-A2.7B | -           | -              | 47.10      |
> | LocFT-BF          | 96.30       | 40.70          | 45.56      |
>
> [1] The Mirage of Model Editing: Revisiting Evaluation in the Wild. ACL 2025.

---

> ### Author Response · Authors · 2025-11-18
> **Response to Reviewer QAtd (2/2)**
>
> **W #4: searching the optimal tuning location**
>
> The extensive fine-tuning experiments across all layers and candidate modules were conducted not merely to search for an optimal location, but to **provide a systematic analysis of how different modules behave** under localized fine-tuning, an aspect missing in existing model editing literature.
>
> The analysis indicates that **module choice has only a modest impact on editing performance**: even without searching for an optimal location, the corrected localized fine-tuning already yields strong results across most modules. This further highlights that the core contribution of our work is correcting the flawed DF pipeline, rather than relying on module-specific optimization.
>
> As demonstrated in `W #3`, in practical applications, a simple default choice within later-layer MLP modules performs robustly on unseen dataset and MoE model, eliminating the need for exhaustive location search.
>
> **Q #1: process strategy for new samples**
>
> When new editing samples arrive, LocFT-BF simply **integrates them incrementally** and performs a new round of fine-tuning. As discussed in `W #1`, the fine-tuning step in LocFT-BF is sufficiently lightweight to make this strategy practical in real-world scenarios. Moreover, in practical applications, new edits can be accumulated into small batches before triggering the next update, which further reduces the computational overhead.
>
> Conflicting edits are a common challenge in sequential editing paradigms [1], where different answers may be injected for the same edit prompt at different times, leading to inconsistencies in the model's outputs. However, the incremental fine-tuning approach provides a natural solution: **conflicts can be detected and resolved at the data level** (e.g., through deduplication) before performing the next update, ensuring that the injected knowledge remains **consistent and coherent**.
>
> [1] Unveiling the Pitfalls of Knowledge Editing for Large Language Models. ICLR 2024.
>
> **Q #3: comparison of cost between BF and DF pipelines**
>
> To provide a fair comparison of the time and computational cost for the BF and DF pipelines, we apply both approaches to edit 1K samples from ZsRE across three LLMs under **identical hyperparameter settings** (batch size = 1, learning rate, loss threshold, etc.). As shown in the following tables, the BF pipeline incurs **negligible additional memory overhead** compared to DF. While BF requires more time per edit, the overall cost remains modest because fine-tuning is inherently efficient.
>
> | GPU Memory (GB) | LLaMA3-8B | Qwen2.5-7B | Mistral-7B |
> | --------------- | --------- | ---------- | ---------- |
> | DF              | 32        | 31         | 29         |
> | BF              | 34        | 31         | 29         |
>
> | Time per Sample (s) | LLaMA3-8B | Qwen2.5-7B | Mistral-7B |
> | ------------------- | --------- | ---------- | ---------- |
> | DF                  | 0.385     | 0.395      | 0.472      |
> | BF                  | 0.720     | 0.672      | 0.754      |
>
> We hope these clarifications have fully addressed your concerns. If you have any further questions, please feel free to reach out.

---

> > ### Comment · Reviewer_QAtd · 2025-11-21
> >
> > Thanks for the detailed response.  I have raised my score to 6. Exploring the reasons behind the success of these techniques and providing intuitive explanations would contribute to the overall scientific contribution of the work.

---

> > > ### Author Response · Authors · 2025-11-22
> > > **Thank you for your thoughtful follow-up.**
> > >
> > > Dear Reviewer QAtd,
> > >
> > > Thank you very much for your thoughtful follow-up and for raising your score.
> > >
> > > We are glad to hear that our responses have addressed your concerns. We deeply appreciate your encouraging feedback and your suggestion to further explore the underlying reasons behind the success of localized fine-tuning. We agree that developing more intuitive explanations would meaningfully strengthen the scientific contribution. This is indeed a direction we have already been planning to explore, and we will further pursue it as part of our future work.
> > >
> > > If there are any remaining aspects you believe could further improve the paper, we would be more than happy to continue the discussion. Thank you again for your time, support, and constructive feedback.
> > >
> > > Best regards,
> > >
> > > Authors

---

### Author Response · Authors · 2025-12-02
**Summary of the Discussion Phase**

Dear Reviewers, AC, SAC, and PCs:

We sincerely appreciate your time and dedication to the ICLR community and the peer-review process. To assist your assessment, we provide a concise summary of (i) the paper's core contributions, (ii) our responses to main concerns and corresponding PDF revisions, and (iii) the review–rebuttal outcome reached by November 25 (**final ratings: 6, 6, 8, 8**).

---

**Contributions and Reviewer Recognition**

We summarize the core contributions and the corresponding feedback from reviewers:

- **Correcting the Misconception regarding Fine-Tuning.**

  - We demonstrate that prior failures stemmed from the Depth-First pipeline; restoring the Breadth-First pipeline reveals fine-tuning is inherently effective for model editing.

  - **Reviewer Feedback:** Reviewers recognized this as an "original proposal" (`Dcsj`) with "rigorous" (`EfTQ`) and "thorough" (`Dcsj`) analysis, finding the core insight "indeed convincing" (`QAtd`) and "exceptionally well-supported" (`MVzX`). They further noted that the work **"overturns the long-standing misconception"** (`EfTQ`) with the potential to **"reshape the model editing field"** (`Dcsj`).

- **Setting New Standards: 72B Models & 100K Edits.**

  - We provide the first demonstration of reliable editing on 72B models and 100K edits, **scaling 10x beyond prior practice**.

  - **Reviewer Feedback:** Reviewers highlighted that the work "demonstrates unprecedented scalability" and "significantly outperforms existing methods" (`EfTQ`), emphasizing that the results meet "real-world deployment needs" and set **"a new bar for evaluation standards"** (`Dcsj`).

- **Simplicity as Strength.**

  - In contrast to complex specialized editing methods, we demonstrate that the **simple and lightweight** fine-tuning approach consistently achieves SOTA performance.

  - **Reviewer Feedback:** Reviewers agreed that LocFT-BF is "simple and efficient" (`MVzX`, `EfTQ`), "easy to implement" (`MVzX`), and "lightweight, plug-and-play, and avoids architectural overheads" (`Dcsj`), establishing a **"highly practical and powerful new baseline"** (`MVzX`).

---

**Responses to Main Concerns and Manuscript Revisions**

During the discussion phase, we actively engaged with all reviewers, addressed their concerns, and incorporated their valuable suggestions into the revised manuscript (updated PDF). Below, we summarize our responses to shared main concerns, while specific minor points are resolved in the respective reviewer threads.

- **Universality of LocFT-BF and its layer-selection strategy.**  (`QAtd`: `W3`, `W4`; `EfTQ`: `W4`; `Dcsj`: `W1`, `W3`)
  - We validate LocFT-BF with the default layer-selection strategy on `Qwen1.5-MoE-A2.7B` (`Appendix A.2`) and a medical editing benchmark `MedEditBench` (`Appendix A.3`), demonstrating superior universality across **heterogeneous architectures** and **specialized domains**.

- **Generalization gap on the CounterFact dataset.**  (`EfTQ`: `W3`; `MVzX`: `W1`; `Dcsj`: `W2`)
  - We show that incorporating the same data augmentation strategy used by baselines yields **substantial generalization gains** for LocFT-BF, enabling it to surpass the leading methods (`Appendix A.4`).

- **Handling asynchronous edits and its efficiency.**  (`QAtd`: `W1`, `Q1`; `Dcsj`: `Q5`)
  - We clarify that LocFT-BF naturally accommodates asynchronous edits by incrementally integrating new samples, and its lightweight fine-tuning procedure ensures that updates remain highly efficient (`Section 2.5`).

- **Scalability of baselines.**  (`EfTQ`: `W5`)
  - We demonstrate that existing baselines do not scale well to 14B models, with the most competitive methods exhibiting substantial performance degradation (`Appendix A.5`).

---

**Rebuttal Outcome**

All reviewers explicitly confirmed that their concerns were resolved, as detailed in the table below.

| Reviewer | **Reviewer's Verification**                                  | Final Rating                                            | Decision Date |
| -------- | -------------------------------------- | -------------------------- | ----- |
| **QAtd** | "Thanks for the detailed response. I have **raised my score to 6**..." | **6**| Nov 21|
| **EfTQ** | "Thank you for adopting my review comments... I have **raised my score to 6** and wish you all the best." | **6** (Raised soundness & contribution to **3** "good") | Nov 19|
| **MVzX** | "I will maintain my positive score and have **increased my confidence rating**." | **8** (Raised confidence to **4**)                      | Nov 25|
| **Dcsj** | "Overall, I acknowledge the quality of this work, and will **maintain my positive score**." | **8**                                                   | Nov 22|

---

We respectfully submit this summary for your consideration. Thank you again for your time and efforts on our work.

Best regards,

Authors

---

### Meta-Review · Area_Chair_RQt8 · 2025-12-31

**Summary:**

This paper challenges the classical sequential (Depth-First) pipeline for model editing, in which edits are made via fine-tuning in the order of arrival, and instead proposes collecting edits in a dataset and applying fine-tuning in a Breadth-First fashion. In practice, this means that instead of applying sequential fine-tuning for editing, edits are applied offline by sampling i.i.d. from the dataset, mitigating the issues of sequential learning (forgetting). They also propose fine-tuning specific layers and modules through exhaustive search.
The approach is simple and quite effective in practice.

I believe this is more of a strong baseline than a method that definitively solves the editing problem, which is important for re-evaluating progress in the field. Based on this and taking into consideration reviewers' comments, I believe the paper should be accepted.

**Reviewer Concerns:**

Concerns from the  reviewers mainly revolve around:
- **[resolved]** Computational cost, which grows linearly with the number of accumulated edits: simple finetuning remains computationally simpler than more complex localization-based methods.
- **[resolved]** No theoretical justification and analysis of the method: I think this is pretty straightforward, as they avoid the sequential editing, which results in biased updates, a classical problem in continual learning.
- **[resolved]** Claims of data augmentation mitigating generalization degradation are not well supported: added experiments in support.

- The generality of the layer selection and knowledge localization method:
The authors "provide a systematic analysis of how different modules behave under localized fine-tuning". Based on this exhaustive search, they propose a heuristic to choose which layer and module to update. There is no evidence that the method is general.

- How to process new edits asynchronously: they claim that the dataset can be expanded as more edits arrive and fine-tuning can be performed again, but they don't provide any additional experiment for this.

**Reviewer Scores:**

- QAtd: 4 -> 6
- EfTQ: 4 -> 6
- MVzX: 8
- Dcsj: 8

---

### Decision · Program_Chairs · 2026-01-26

Accept (Poster)